# Ongoing resolution of duplicate gene functions shapes the diversification of a metabolic network

**Meihua Christina Kuang**[1,2,3,4,5]**, Paul D Hutchins**[5,6,7]**, Jason D Russell**[5,7,8]**, Joshua J Coon**[5,6,7,8,9]**, Chris Todd Hittinger**[1,2,3,4,5,7]*****

[1]Laboratory of Genetics, University of Wisconsin-Madison, Madison, United States; [2]Graduate Program in Cellular and Molecular Biology, University of Wisconsin-Madison, Madison, United States; [3]Wisconsin Energy Institute, University of Wisconsin-Madison, Madison, United States; [4]JF Crow Institute for the Study of Evolution, University of Wisconsin-Madison, Madison, Madison, United States; [5]Genome Center of Wisconsin, University of Wisconsin-Madison, Madison, United States; [6]Department of Chemistry, University of Wisconsin-Madison, Madison, United States; [7]DOE Great Lakes Bioenergy Research Center, University of Wisconsin-Madison, Madison, United States; [8]Metabolism Research Group, Morgridge Institute for Research, Madison, United States; [9]Department of Biomolecular Chemistry, University of Wisconsin-Madison, Madison, United States

**\*For correspondence:**
cthittinger@wisc.edu

**Competing interests:** The authors declare that no competing interests exist.

**Abstract** The evolutionary mechanisms leading to duplicate gene retention are well understood, but the long-term impacts of paralog differentiation on the regulation of metabolism remain underappreciated. Here we experimentally dissect the functions of two pairs of ancient paralogs of the *GAL*actose sugar utilization network in two yeast species. We show that the *Saccharomyces uvarum* network is more active, even as over-induction is prevented by a second co-repressor that the model yeast *Saccharomyces cerevisiae* lacks. Surprisingly, removal of this repression system leads to a strong growth arrest, likely due to overly rapid galactose catabolism and metabolic overload. Alternative sugars, such as fructose, circumvent metabolic control systems and exacerbate this phenotype. We further show that *S. cerevisiae* experiences homologous metabolic constraints that are subtler due to how the paralogs have diversified. These results show how the functional differentiation of paralogs continues to shape regulatory network architectures and metabolic strategies long after initial preservation.

## Introduction

Gene duplication provides raw material for evolution to act upon. Even so, most duplicate genes are inactivated and become pseudogenes before fixation. The molecular mechanisms behind paralog retention and differentiation have attracted considerable attention, and several general models have been proposed, including neofunctionalization (*Ohno, 1970*; *Zhang et al., 2002*), gene dosage selection (*Conant and Wolfe, 2007*; *Sandegren and Andersson, 2009*; *Conant et al., 2014*), subfunctionalization by duplication-degeneration-complementation (*Force et al., 1999*), and subfunctionalization by escape from adaptive conflict (*Hittinger and Carroll, 2007*; *Des Marais and Rausher, 2008*). Theoretical studies have proposed that the fates of duplicate genes are rapidly determined after gene duplication events (*Moore and Purugganan, 2003*; *Innan and Kondrashov, 2010*). These models generally treat the preservation of duplicate genes as a race to distinguish

**eLife digest** Genetic information is organized into units called genes, which encode sets of instructions needed to make proteins and other molecules in cells. When an organism reproduces, it passes on some or all of its genes to its offspring. Over many generations, individual genes may acquire changes known as mutations. Some mutations may improve the ability of the individual to survive and reproduce, but most are harmful and may even lead to death.

Organisms can bypass these constraints by creating extra copies of genes so that if one copy acquires a harmful mutation the other copy still works normally. Duplicate genes are crucial to evolution because they offer opportunities to evolve new characteristics, but most duplicated genes are quickly inactivated, in part because the organism does not need them. Decades of research have focused on how some duplicate genes manage to remain active, often by specializing or acquiring new roles. However, relatively little attention has been paid to what happens to duplicates after they have been retained.

Baker's yeast – also known as *Saccharomyces cerevisiae* – is a single-celled organism that is often used to study genetics. Kuang et al. investigated whether yeast genes that were duplicated millions of years ago are still actively evolving and whether they have achieved different evolutionary outcomes in baker's yeast and a closely related yeast called *Saccharomyces uvarum*. The experiments examined two pairs of genes that help to break down a sugar called galactose. Even though these genes were duplicated around 100 million years ago, they have continued to evolve in these yeasts.

In the last ten million years, these duplicate genes have taken on different roles so that baker's yeast and *S. uvarum* have evolved different strategies for consuming galactose. *S. uvarum* breaks down galactose more aggressively than baker's yeast, but it also has a better system in place to prevent it from breaking down more galactose than it needs.

The findings of Kuang et al. suggest that the new roles that duplicated genes adopt in organisms might have a bigger effect on the evolution of organisms in the long run than is currently appreciated. Future work will test this idea by studying other duplicated genes in different species.

their functions prior to the complete inactivation of one of the redundant paralogs, either through neutral (*Force et al., 1999*; *Lynch et al., 2001*) or adaptive changes (*Clark, 1994*; *Lynch et al., 2001*). Regardless of the initial functional changes or dosage effects facilitating the fixation of paralogs, retention is not the end of their evolutionary paths (*Gordon et al., 2009*; *Conant et al., 2014*).

Duplicate genes continue to diverge in different lineages, providing additional evolutionary opportunities for organisms to diversify. Previously fixed copies of duplicate genes can alter their expression timing and patterns (*Huminiecki and Wolfe, 2004*; *Tümpel et al., 2006*), change substrate affinities (*Voordeckers et al., 2012*), and switch between regulatory modules (*Thompson et al., 2013*). In several cases, paralogs encoding enzymes have been recruited to perform regulatory functions, such as *S. cerevisiae HXK2*, *GAL3*, and *ARG82* (*Gancedo and Flores, 2008*; *Conant et al., 2014*; *Gancedo et al., 2014*). Previously differentiated developmental roles can even be transferred from one paralog to another during evolution (*Ureña et al., 2016*). Perhaps more significantly, long-preserved paralogs can be lost in lineage-specific manners, a common phenomenon observed across the tree of life, including in bacteria (*Gómez-Valero et al., 2007*), yeasts (*Scannell et al., 2007*), *Paramecium* (*Aury et al., 2006*; *McGrath et al., 2014*), plants (*De Smet et al., 2013*), fishes (*Amores et al., 2004*), and mammals (*Amores et al. 1998*; *Blomme et al., 2006*). Although pervasive, the importance of ongoing paralog diversification to the evolution of organismal traits and phenotypes remains underappreciated.

Duplicate gene differentiation has heavily impacted the evolution of regulatory and metabolic networks (*Reece-Hoyes et al., 2013*; *Voordeckers et al., 2015*). Paralogs have contributed to the expansion of regulatory networks (*Teichmann and Babu, 2004*), the derivation of novel networks (*Conant and Wolfe, 2006*; *Wapinski et al., 2010*; *Pérez et al., 2014*; *Pougach et al., 2014*), the specialization of network regulation (*Lin and Li, 2011*), and the robustness of networks to perturbation (*Papp et al., 2004*; *Deutscher et al., 2006*). The WGD has even been proposed to have

facilitated the evolution of an aerobic glucose fermentation strategy called Crabtree-Warburg Effect in the lineage of yeasts that includes *Saccharomyces* (*Conant and Wolfe, 2007*; *Jiang et al., 2008*). Gene regulation and metabolism are heavily intertwined biological processes, but there are few eukaryotic models that allow for an integrated study of the ongoing differentiation of paralogous genes with regulatory and metabolic diversification (*Yamada and Bork, 2009*; *Conant et al., 2014*).

The *Saccharomyces cerevisiae GAL*actose sugar utilization network is one of the most extensively studied eukaryotic regulatory and metabolic networks, and its homologous networks are evolutionarily dynamic in yeasts. In *S. cerevisiae*, it includes the three enzymes of the Leloir pathway (Gal1, Gal7, and Gal10) that catabolize galactose, the galactose transporter Gal2, and three regulators. In the absence of galactose, the transcription factor Gal4 is inhibited by the co-repressor Gal80. When galactose is present, Gal80 is sequestered by the co-inducer Gal3, allowing Gal4 to activate the expression of the *GAL* network (*Johnston, 1987*; *Bhat and Murthy, 2001*; *Egriboz et al., 2013*). Numerous studies have shown that the *GAL* networks of various yeast lineages vary in gene content (*Hittinger et al., 2004*, *2010*; *Wolfe et al., 2015*) and gene activity (*Peng et al., 2015*; *Roop et al., 2016*). Despite these findings, the impacts of variable network architectures on the evolution of gene regulation and metabolism are not well understood.

As a model for how duplicate gene divergence creates variable network architectures, we functionally characterized the *GAL* network of *Saccharomyces uvarum* (formerly known as *Saccharomyces bayanus* var. *uvarum*) and compared it to *S. cerevisiae*. Here we show that two *GAL* network paralog pairs in *S. uvarum* have diverged to different degrees and states than their *S. cerevisiae* homologs. We further show that, unlike *S. cerevisiae*, *S. uvarum* deploys a second co-repressor that prevents over-induction of the network. *S. uvarum* mutants lacking both co-repressors revealed surprising constraints on the rapid utilization of galactose; specifically, they arrested their growth, and metabolomic investigations suggested that they experienced metabolic overload. We show that homologous constraints exist in a milder form in *S. cerevisiae*, and the degree of metabolic constraint is affected by how *GAL* network paralogs have diversified between the species. These results show how, after a hundred of million of years of preservation, two pairs of interacting duplicate genes have continued to diverge functionally in ways that broadly impact metabolism, regulatory network structures, and the future evolutionary trajectories available.

## Results

### GAL gene content and sequence differences

*S. uvarum* has orthologs encoding all regulatory and structural genes that are present in *S. cerevisiae*, but it has duplicate copies of two additional genes. The first additional duplicate gene is *GAL80B*, which is a paralog of *GAL80*; this pair of paralogs was created by the whole genome duplication (WGD) event roughly 100 million years ago (*Wolfe and Shields, 1997*; *Marcet-Houben and Gabaldón, 2015*). *GAL80B* has been retained in the *S. uvarum-Saccharomyces eubayanus* clade, but it was lost in the *S. cerevisiae-Saccharomyces arboricola* clade (*Hittinger et al., 2010*, *2004*; *Scannell et al., 2011*; *Caudy et al., 2013*; *Hittinger, 2013*; *Liti et al., 2013*; *Baker et al., 2015*). The second one is *GAL2B*, which was created by a recent tandem duplication in *S. uvarum-S. eubayanus* clade. Both *S. cerevisiae* and *S. uvarum* also contain a pair of specialized paralogs created by the WGD, *GAL1* and *GAL3*. By comparing amino acid sequences against the *S. cerevisiae GAL* network, we found that most *GAL* genes are diverged to a similar extent (*Figure 1* and *Figure 1—source data 1*), except for *GAL4*, which is primarily conserved in its DNA-binding and other functionally characterized domains. None of the *S. uvarum GAL* homologs exhibited significantly elevated rates of protein sequence evolution (from previously calculated $d_N/d_S$ ratios [*Byrne and Wolfe, 2005*]), which might have otherwise suggested extensive neofunctionalization. Thus, we focused on whether and how the key regulatory genes partitioned functions differently between the two species.

### Less partitioned galactokinase and co-induction functions

In *S. cerevisiae*, the *GAL1* and *GAL3* paralogs are descended from an ancestral bi-functional protein that was both a co-inducer and a galactokinase (*Rubio-Texeira, 2005*; *Hittinger and Carroll, 2007*). They are almost completely subfunctionalized: ScerGAL3 lost its galactokinase activity and became a

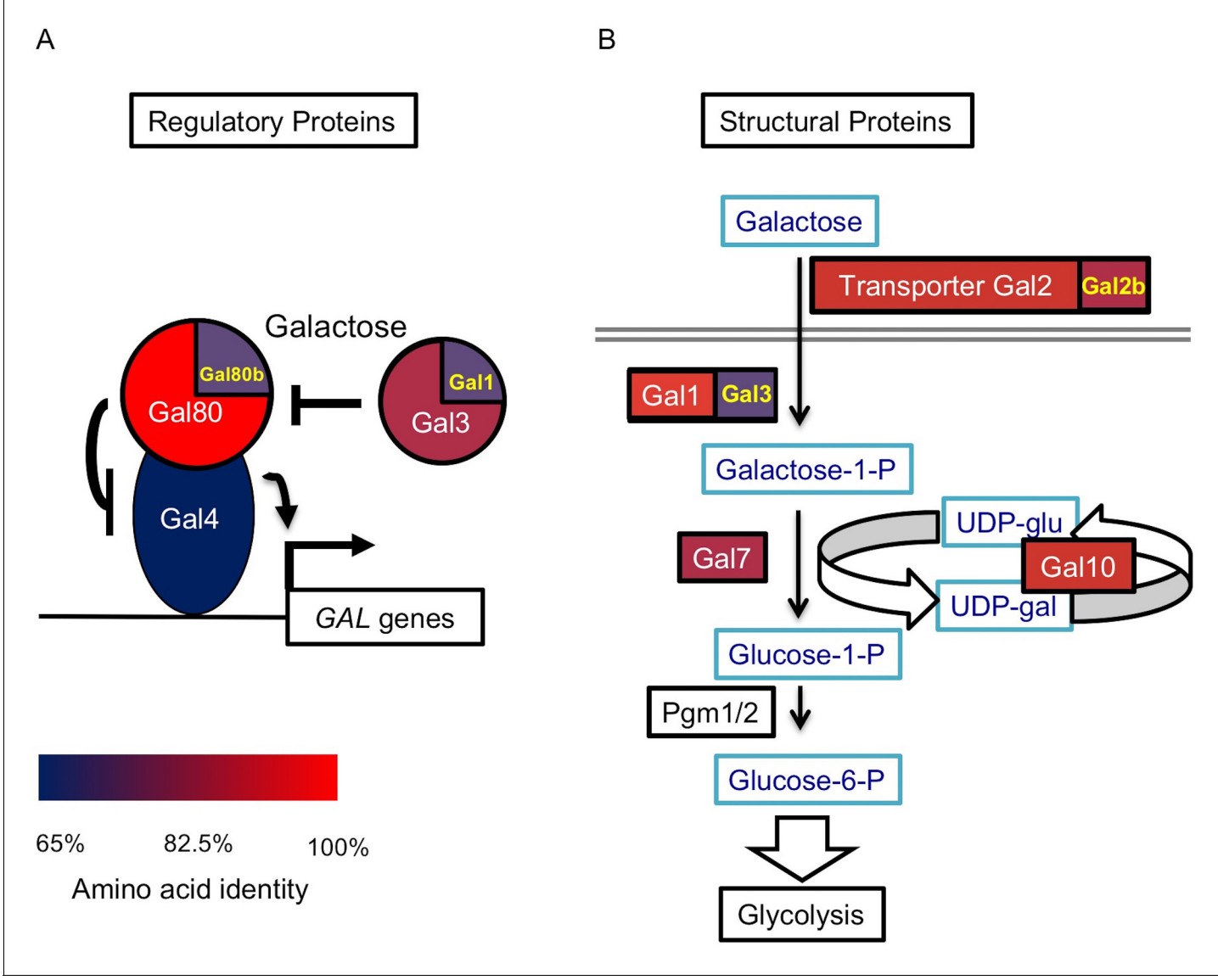

**Figure 1.** The *S. uvarum GAL* network. (A) The *GAL* regulatory network. (B) The *GAL* or Leloir metabolic pathway. The colors show the amino acid identity of each component compared to their *S. cerevisiae* homologs (full data in *Figure 1—source data 1*). Proteins with two homologs in *S. uvarum* are split into two parts: Gal1/Gal3 and Gal80/Gal80b (also known as *Sbay_12.142* [*Scannell et al., 2011*] or *670.20* [*Caudy et al., 2013*]) are two pairs of paralogs from a WGD event, while Gal2/Gal2b (also known as *Sbay_10.165* [*Scannell et al., 2011*] or *672.62* [*Caudy et al., 2013*]) are paralogs from a recent tandem duplication event (*Hittinger et al., 2004*).

The following source data is available for figure 1:

**Source data 1.** Amino acid identity and *GAL* gene composition between *S. uvarum* and *S. cerevisiae GAL* network.

dedicated co-inducer, whereas ScerGAL1 lost most of its co-inducer activity but maintains galactokinase activity (*Platt and Reece, 1998*; *Platt et al., 2000*; *Timson et al., 2002*; *Hittinger and Carroll, 2007*; *Lavy et al., 2016*). Unlike ScerGal3, SuvaGal3 retains a -Ser-Ala- dipeptide in its active site that is sufficient to weakly restore galactokinase activity when added back to ScerGal3 (*Platt et al., 2000*), so we hypothesized that *SuvaGAL3* encodes a functional galactokinase. To test this hypothesis, we precisely replaced the coding sequence of *ScerGAL1*, the gene encoding the sole galactokinase in *S. cerevisiae* (*Platt et al., 2000*), with *SuvaGAL3* in *S. cerevisiae*. As expected, *SuvaGAL3* conferred robust growth in galactose when driven by the *ScerGAL1* promoter, suggesting that

SuvaGAL3 retains galactokinase activity (*Figure 2A*). Nonetheless, the *S. uvarum gal1* null mutant did not grow better in 2% galactose than it did without any carbon source, a phenotype similar to the *S. cerevisiae gal1* null mutant (*Figure 2—figure supplement 1*), indicating that the native *GAL3* promoter expression is insufficient to support robust metabolism.

To further examine the functional divergence between *SuvaGAL1* and *SuvaGAL3*, we knocked out *GAL3* in *S. uvarum*. Surprisingly, the *S. uvarum gal3* null mutant did not show the classic Long-Term Adaptation (LTA) phenotype of the *S. cerevisiae gal3* null mutant (*Tsuyumu and Adams, 1973*). Instead of a growth delay of multiple days, we observed a delay of only a few hours in *S. uvarum gal3Δ* relative to wild-type (*Figure 2B*). These results suggest that other genes in *S. uvarum* may be able to partially compensate for the deletion of *SuvaGAL3*, such as its paralog, *SuvaGAL1*. To determine whether *GAL1* differences between *S. uvarum* and *S. cerevisiae* might be responsible for the different *gal3* null phenotypes, we replaced the *SuvaGAL1* coding sequence or promoter sequence with their *ScerGAL1* counterparts in the background of *S. uvarum gal3Δ*. The *ScerGAL1* promoter swap in *S. uvarum gal3Δ* largely recapitulated LTA, while the *ScerGAL1* coding sequence swap extended the delay to a lesser extent (*Figure 2C*). Since the *GAL1-GAL10* promoter is a divergent promoter, genetic modifications (evolved or engineered) inevitably impact both genes, as well as perhaps a lncRNA previously described in *S. cerevisiae* (*Cloutier et al., 2016*). These results suggest that differences at the *GAL1* locus, especially within this promoter, are primarily responsible for the lack of LTA in the *S. uvarum gal3Δ* mutant. Overall, the data suggest that *SuvaGAL1* is functionally redundant with *SuvaGAL3* to a much greater extent than are *ScerGAL1* and *ScerGAL3*. Thus, it is likely that the homologs in the common ancestor of *S. uvarum* and *S. cerevisiae* were more functionally redundant than in modern *S. cerevisiae*, and considerable subfunctionalization between *ScerGAL1* and *ScerGAL3* happened after the divergence of *S. uvarum* and *S. cerevisiae*.

## *S. uvarum* has two co-repressors with partially overlapping functions

Next, we examined the functional divergence of the other pair of paralogous regulatory genes, *SuvaGAL80* and *SuvaGAL80B*, which are homologous to the *ScerGAL80* gene that encodes the sole *GAL* gene co-repressor in *S. cerevisiae*. We first examined the expression of these two genes in the presence or absence of galactose (*Figure 3A*). RNA sequencing (RNA-Seq) showed that *SuvaGAL80* was expressed at a higher level than *SuvaGAL80B* in the absence of galactose (i.e. with glycerol or glucose as the sole carbon source). In contrast, in the presence of galactose, *SuvaGAL80B* was induced by 133-fold, much higher than the 6-fold induction observed for *SuvaGAL80* (*Figure 3B*). *S. uvarum gal80* null mutants had a shorter lag time than wild-type in galactose, as seen in *S. cerevisiae gal80* null mutants (*Torchia et al., 1984*; *Segrè et al., 2006*; *Hittinger et al., 2010*), but *gal80b* null mutants did not (*Figure 3C*). Deleting *SuvaGAL80* also resulted in elevated *GAL1* expression in the non-inducing condition (i.e. 5% glycerol), while deleting *SuvaGAL80B* had no detectable effect (*Figure 3D*). Therefore, we conclude that *SuvaGAL80* is the main gene responsible for repressing the *GAL* network in the absence of galactose.

Perhaps because of its dynamic expression, the deletion mutant phenotype of *S. uvarum gal80bΔ* proved condition dependent. Consistent with previous negative results (*Caudy et al., 2013*), no apparent phenotypic differences were observed for the *S. uvarum gal80bΔ* strain when it was grown in galactose, nor were its *GAL1* expression levels altered (*Figure 3C* and *Figure 3—figure supplement 1*). Nonetheless, in a mixture of galactose and glucose, we observed elevated *GAL1* expression in *S. uvarum gal80bΔ* strains, beyond the levels observed in *S. uvarum gal80Δ* strains (*Figure 3E*). Additionally, *S. uvarum gal80bΔ* grew significantly slower than wild-type after transfer from galactose to a mixture of galactose and glucose (*Figure 3F*), suggesting that *SuvaGAL80B* plays a specific and biologically important repressive role in conditions where it is required to prevent network over-induction. We also observed strong negative epistasis when both co-repressors were removed: the co-repressor double knockout had substantially higher *GAL1* expression than either single knockout strain or the *S. uvarum* wild-type strain in the absence of galactose (*Figure 3G*). Thus, *SuvaGAL80* and *SuvaGAL80B* encode partially redundant *GAL* gene co-repressors that can each partially compensate for the loss of the other. We conclude that *SuvaGAL80B* may play a minor role in the absence of galactose, but it provides important modulation in induced conditions.

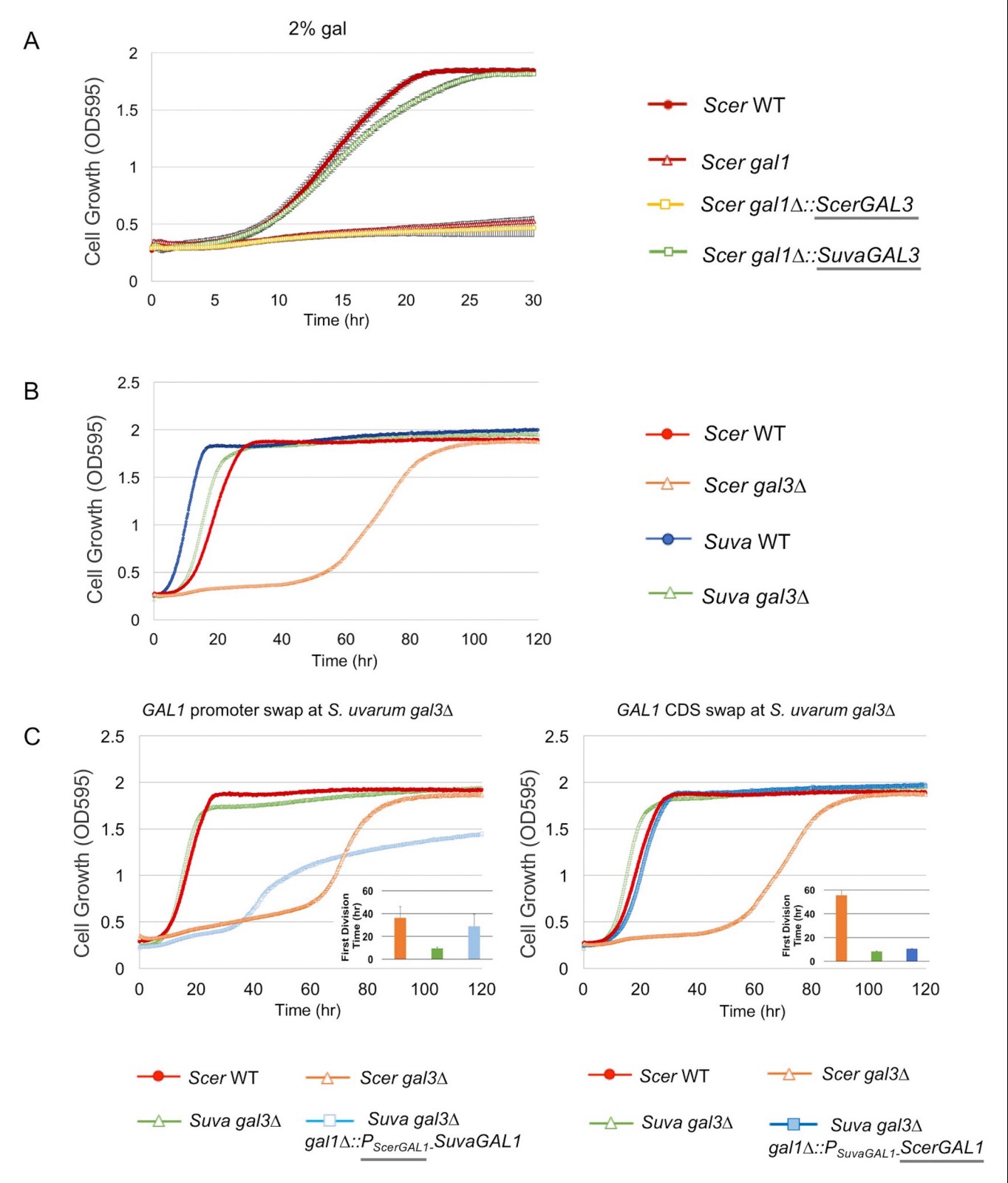

**Figure 2.** *SuvaGAL1* and *SuvaGAL3* are not as subfunctionalized as *ScerGAL1* and *ScerGAL3*. (**A**) *S. uvarum GAL3* likely encodes a functional galactokinase. The error bars represent standard deviations of three biological replicates. A Wilcoxon rank sum test comparing the average times to first doubling between *S. cerevisiae gal1* and *S. cerevisiae gal1Δ::SuvaGAL3* was significantly different (p=5.2e-3, *n* = 6). Note that driving *ScerGAL3* from the *ScerGAL1* promoter was insufficient to support growth with galactose as the sole carbon source, but *SuvaGAL3* was sufficient. (**B**) Unlike *S.*
*Figure 2 continued on next page*

*Figure 2 continued*

*cerevisiae gal3Δ, S. uvarum gal3Δ* does not show Long-Term Adaption (LTA). Strains were cultured in SC + 2% galactose. Wilcoxon rank sum tests comparing the average times to first doubling between *S. uvarum gal3Δ* and *S. uvarum* wild-type strains were significantly different (p=4.5e-5, *n* = 12). These experiments have been repeated independently at least twice with three biological replicates, but growth curves display only one representative replicate because LTA emergence is stochastic. (C) LTA was recapitulated in *S. uvarum gal3Δ* by replacing its *GAL1* promoter with the *S. cerevisiae GAL1* promoter (left panel) or, to a much lesser extent, by replacing the coding sequence (right panel). The insets show the times to the first doubling for the strains for their respective panels. The bar colors in the inset are the same as the growth curves. To highlight strain comparisons that test discrete hypotheses, three genotypes are repeated in *Figure 2B* and in both panels of *Figure 2C*: *S. uvarum gal3Δ, S. cerevisiae* wild-type, and *S. cerevisiae gal3Δ*. Strains were cultured in SC + 2% galactose. Wilcoxon rank sum tests comparing the average times to first doubling between strains were as follows: (1) p=4.6e-4 and *n* = 9 for *S. uvarum gal3Δ* versus *S. uvarum gal3Δ gal1Δ::P_{SuvaGAL1}-ScerGAL1*, (2) p=4.2e-5 and *n* = 12 for *S. uvarum gal3Δ* versus *S. uvarum gal3Δ gal1Δ::P_{ScerGAL1}-SuvaGAL1*, and (3) p=0.21 and *n* = 12 for *S. uvarum gal3Δ gal1Δ::P_{ScerGAL1}-SuvaGAL1* versus *S. cerevisiae gal3*.

The following figure supplement is available for figure 2:

**Figure supplement 1.** *S. uvarum* and *S. cerevisiae* have qualitatively similar *gal1* null phenotypes.

## Strains lacking the co-repressors arrest their growth

Surprisingly, knocking out both *GAL80* and *GAL80B* in *S. uvarum* resulted in a strong Temporary Growth Arrest (TGA) phenotype in galactose (*Figure 4A*). This result stands in sharp contrast to the observation that *S. cerevisiae gal80* null mutant strains from multiple genetic backgrounds (the lab strains S288c, W303, and R21, as well as the vineyard strain RM11-1a examined here) grew faster in galactose, a phenotype shared with *Saccharomyces kudriavzevii gal80* null mutants and attributed to the constitutive *GAL* expression (*Torchia et al., 1984*; *Segrè et al., 2006*; *Hittinger et al., 2010*). This growth arrest was not a genetic engineering artifact; reintroducing *SuvaGAL80* completely rescued the growth arrest, and knocking out these two genes with different markers produced the same mutant phenotype (*Figure 4—figure supplement 1*). More importantly, introducing *ScerGAL80* completely rescued the growth arrest (*Figure 4—figure supplement 1*), suggesting that the TGA phenotype was not due to novel molecular functions specific to *SuvaGAL80* or *SuvaGAL80B*. Instead, the dramatically varied phenotypes imply that these two species have different regulatory or metabolic wiring for galactose metabolism.

To test whether the TGA phenotype was associated with *S. uvarum*-specific *GAL* network members, we performed RNA-Seq on *S. uvarum gal80Δ gal80bΔ* in 2% glucose or 5% glycerol, conditions where the complete *GAL* network is expected to be constitutively expressed (*Torchia et al., 1984*; *Segrè et al., 2006*; *Hittinger et al., 2010*). We identified genes as *GAL* network members if and only if they were: (1) significantly up-regulated in *S. uvarum gal80Δ gal80bΔ* compared to the wild-type at FDR = 0.05 (35 genes); (2) up-regulated by at least two-fold (19 genes); (3) up-regulated in both glucose and glycerol (nine genes); and (4) predicted to contain Gal4 consensus binding sites (CGGN$_{11}$CCG) upstream of their coding sequences. Using these stringent criteria, we found eight potential *GAL* network members in *S. uvarum*, seven of which were shared with *S. cerevisiae* based on previous chromatin immunoprecipitation and gene expression data (*GAL1, GAL2, GAL2B, GAL7, GAL10, MEL1*, and *GCY1*) (*Torchia et al., 1984*; *Ren et al., 2000*) (*Figure 4—figure supplement 2A*). *GAL3*, a well-established Gal4 target in *S. cerevisiae*, was considered differentially expressed using less stringent criteria, but orthologs of two other known targets were not (*MTH1* and *PCL10*). The sole novel *GAL* network member in *S. uvarum* was the *PGM1* gene, which was up-regulated 26-fold in 5% glycerol in *S. uvarum gal80Δ gal80bΔ* relative to wild-type. In *S. cerevisiae*, *PGM1* encodes the minor isoform of phosphoglucomutase, which, along with Pgm2, connects the Leloir pathway to glycolysis (*Figure 1*). Notwithstanding the *PGM1* gene, we conclude that the *S. uvarum* and *S. cerevisiae GAL* networks have similar compositions, and the handful of differences do not seem to readily explain the remarkably strong and unexpected TGA phenotype seen in *S. uvarum* strains lacking their co-repressors.

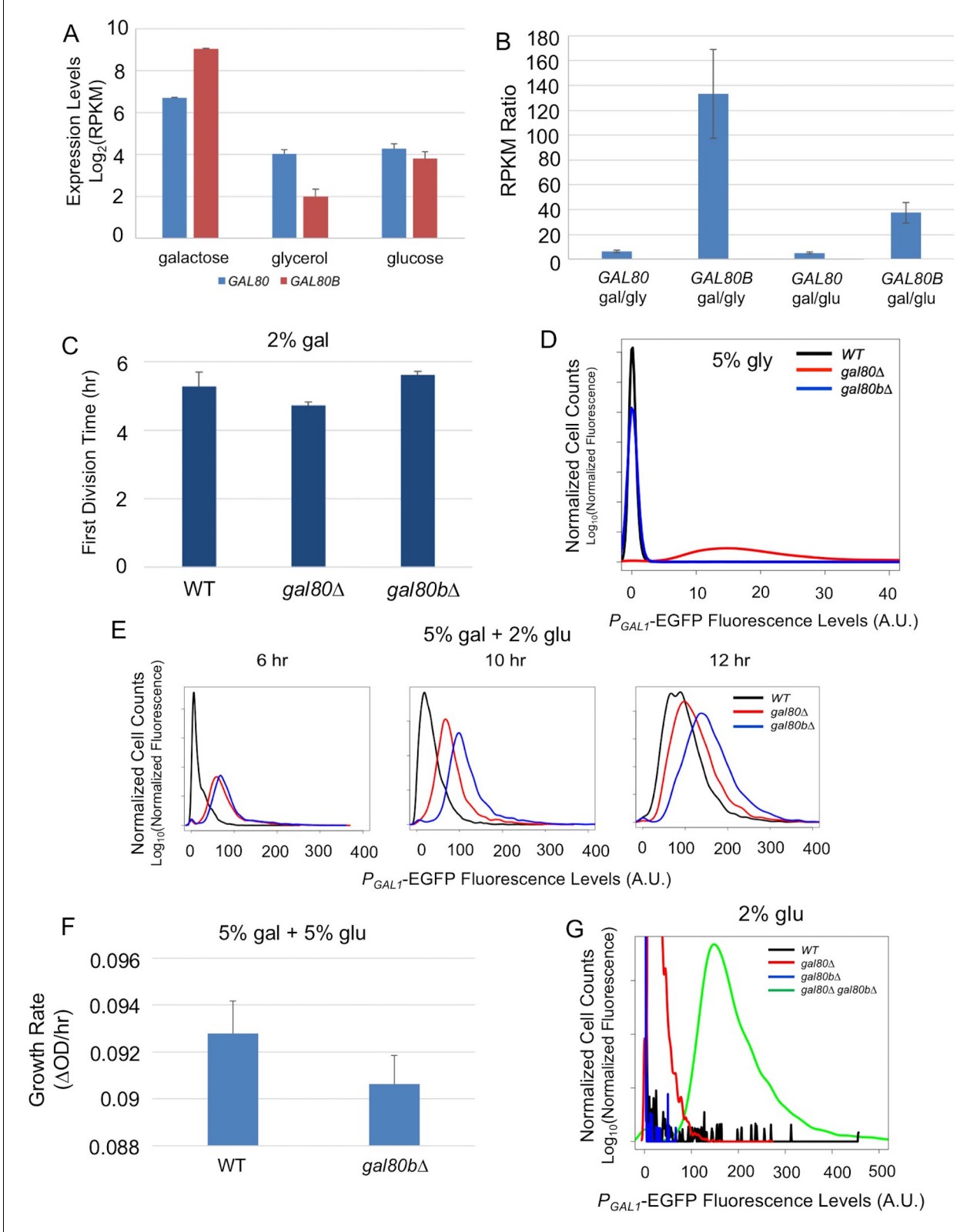

**Figure 3.** *SuvaGAL80* and *SuvaGAL80B* encode co-repressors with partially overlapping functions. (**A**) Expression divergence between *SuvaGAL80* and *SuvaGAL80B*. The bar graph on the left shows the mRNA levels (in log₂ of Reads Per Kilobase of transcript per Million mapped reads or RPKM) of *SuvaGAL80* and *SuvaGAL80B* in SC + 2% galactose, SC + 5% glycerol, and SC + 2% glucose. Error bars represent the standard deviations of three biological replicates. (**B**) Divergent galactose induction between *SuvaGAL80* and *SuvaGAL80B*. The bar graph shows the ratio of mRNA levels between

*Figure 3 continued on next page*

*Figure 3 continued*

galactose (gal) and glycerol (gly), or between galactose and glucose (glu) from the data in Panel A. (**C**) Removing *SuvaGAL80* conferred rapid initial growth in galactose. The bar graph shows the average time to first doubling of three biological replicates of each genotype in SC + 2% galactose from a representative experiment. *S. uvarum gal80Δ* grew significantly faster than wild-type (p=1.8e-3, n = 14, Wilcoxon rank sum test), but *S. uvarum gal80bΔ* did not (p=0.61, n = 14, Wilcoxon rank sum test). (**D**) Removing *SuvaGAL80* resulted in constitutive *GAL1* expression. The histogram shows the fluorescence levels of an EGFP reporter when driven by the *S. uvarum GAL1* promoter in SC + 5% glycerol as determined by flow cytometry. (**E**) Removing *SuvaGAL80B* led to the elevated *GAL1* expression in a mixture of glucose and galactose. Flow cytometry was conducted on strains cultured in SC + 5% galactose +2% glucose. (**F**) Removing *SuvaGAL80B* caused a fitness defect in a mixture of glucose and galactose. The specific growth rate of *S. uvarum gal80bΔ* was significantly lower than wild-type (p=2.7e-4, n = 18, Wilcoxon rank sum test). (**G**) *SuvaGAL80* and *SuvaGAL80B* were both able to partially compensate for the loss of the other in repressing conditions, but the double-knockout resulted in constitutive expression. The histogram reports flow cytometry data from strains cultured in SC + 2% glucose for 9 hr.

The following figure supplement is available for figure 3:

**Figure supplement 1.** In SC + 2% galactose, *S. uvarum gal80Δ* and *gal80bΔ* had *GAL1* expression levels similar to the wild-type at mid-log phase.

## Overactive galactose catabolism precedes widespread metabolic and regulatory defects

In contrast to the constitutive expression of a fairly small network of direct Gal4 targets seen during growth in glucose and glycerol, *S. uvarum gal80Δ gal80bΔ* double mutants experienced global changes in gene expression that were specific to growth in galactose (*Figure 4—figure supplement 2B,C*). During the TGA phase, 1006 genes were differentially expressed in *S. uvarum gal80Δ gal80bΔ* relative to wild-type (620 genes up-regulated and 386 genes down-regulated by at least two-fold with FDR = 0.05 [*Figure 5—source data 1*]). After the mutant resumed growth in galactose, the vast majority (78%, 783 of 1006 genes) of these genes returned to expression levels indistinguishable from wild-type, and Gene Ontology (GO) term analysis showed that most of the biological processes affected during the TGA phase returned to normal (*Supplementary file 1*). The TGA phase gene expression profile was not consistent with a global environmental stress response (e.g. nuclear ribosome biogenesis and rRNA processing were up-regulated) but instead suggested a complex and incoherent integration of the regulatory signals that govern metabolism (*Figure 5—source data 1* and *Supplementary file 1*).

Several lines of evidence suggested that this mis-regulation might be caused by overly rapid galactose catabolism immediately prior to the TGA phase. First, the optical density of the co-repressor double mutant initially increased faster than the wild-type in galactose and only plateaued after about 1.5 hr (*Figure 4A*). Second, during this early growth in galactose, the co-repressor double mutant produced more ethanol than the wild-type (*Figure 4—figure supplement 3*). Third, *GAL1* was also strongly overexpressed in the mutant early during growth in galactose, but *GAL1* expression gradually converged with the wild-type strain as the cells transitioned into the TGA phase (*Figure 4—figure supplement 4*). Finally, the severity of the TGA phenotype depended strongly on galactose concentration (*Figure 4B*), and growth defects were not seen in other carbon sources (*Figure 4—figure supplement 5*).

To further characterize how overly rapid galactose catabolism might lead to the TGA phenotype, we performed metabolomic analyses using mass spectrometry on co-repressor double mutant and wild-type strains cultured in 2% galactose. Prior to the TGA phase, the co-repressor double mutant accumulated galactose-1-phosphate, a known toxic intermediate of galactose metabolism (*Petry and Reichardt, 1998*; *de Jongh et al., 2008*), but this two-fold accumulation (relative to wild-type) seemed unlikely to be sufficient to explain the TGA phenotype. The level of galactose-1-phosphate in *S. uvarum gal80Δ gal80bΔ* returned to normal during the TGA phase (*Figure 5* and *Figure 5—source data 2*) and was not nearly as strong as in *S. cerevisiae gal7Δ* or *gal10Δ* controls (seven- to 11-fold relative to *S. cerevisiae* wild-type) (*Figure 5—figure supplement 1*). Moreover, we did not observe gene expression signatures consistent with the previously described responses to galactose-1-phosphate toxicity (e.g., environmental stress response, unfolded protein response) (*Slepak et al., 2005*; *De-Souza et al., 2014*).

Instead, both transcriptomic and metabolomic analyses revealed broad metabolic defects as bottlenecks developed downstream of the Leloir pathway. During the growth arrest, we observed

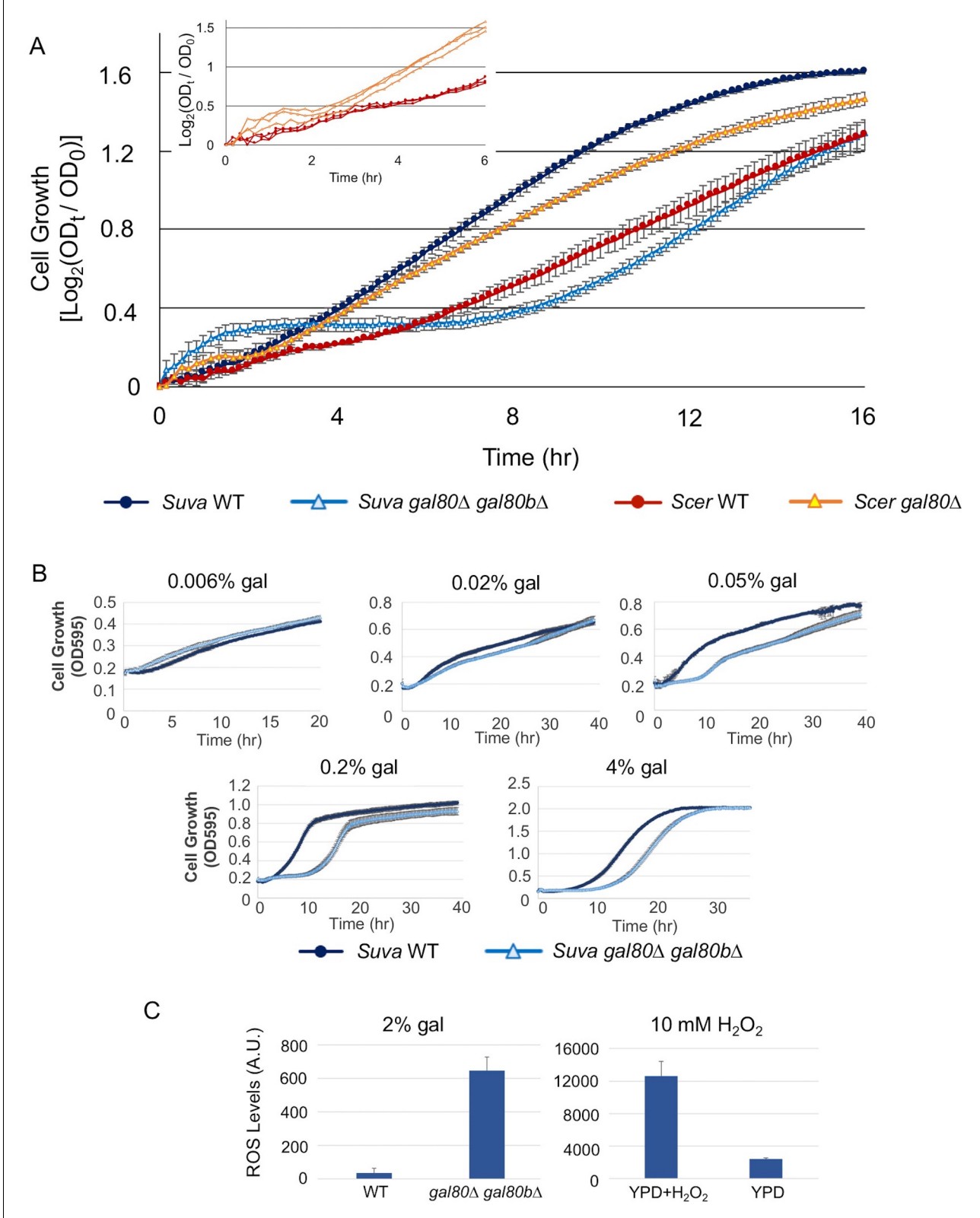

**Figure 4.** The galactose-dependent temporary growth arrest phenotype of *S.uvarum gal80Δ gal80bΔ*. (**A**) The Temporary Growth Arrest (TGA) phenotype in SC + 2% galactose. The averages of the log$_2$ of the ratios between absorbances at each time point (OD$_t$) and initial absorbances (OD$_0$) for three biological replicates are shown. The error bars represent standard deviations. The inset shows the first six hours for three biological replicates each of *S. cerevisiae* wild-type and *gal80Δ* (in the background of *S. cerevisiae* RM11-1a, a vineyard strain). (**B**) The degree of the TGA phenotype was

*Figure 4 continued on next page*

*Figure 4 continued*

concentration dependent. A representative experiment with three biological replicates is shown; the experiment has been repeated three times. (**C**) Excessive reactive oxygen species (ROS) were accumulated in *S. uvarum gal80Δ gal80bΔ* during the TGA phase. ROS levels are reported as relative fluorescence and were measured 6.5 hr after inoculation into SC + 2% galactose (p=8.6e-6, *n* = 11, Wilcoxon rank sum test). The bar graph on the right shows a positive control using *S. uvarum* wild-type in YPD and YPD + 10 mM $H_2O_2$.

The following figure supplements are available for figure 4:

**Figure supplement 1.** The TGA phenotype of *S. uvarum gal80Δ gal80bΔ* can be rescued by *S. cerevisiae GAL80* or by re-introducing *SuvaGAL80*.

**Figure supplement 2.** Galactose-specific global differential expression of *S. uvarum gal80Δ gal80bΔ*.

**Figure supplement 3.** High performance liquid chromatography measurements of key metabolites in SC + 2% galactose in *S. uvarum gal80Δ gal80bΔ* and wild-type during the TGA phase and after the growth resumed.

**Figure supplement 4.** *GAL1* expression was higher at the early stages of growth in SC + 2% galactose in the *S. uvarum gal80Δ gal80bΔ* background but gradually decreased.

**Figure supplement 5.** Fructose, mannose, or glucose alone did not lead to a TGA phenotype or other growth defects.

**Figure supplement 6.** The regulation of *PGM1* by galactose was inferred as the ancestral state.

increased expression of genes that encode glycolytic enzymes (*Figure 5* and *Figure 5—source data 1*). Key metabolic intermediates also accumulated in *S. uvarum gal80Δ gal80bΔ* strains before and during growth arrest, especially in upper glycolysis and interacting pathways (*Figure 5* and *Figure 5—source data 2*). In particular, fructose-1,6-biphosphate accumulated significantly prior to the TGA phase (12.6-fold of wild-type levels) (*Figure 5* and *Figure 5—source data 2*), a bottleneck that frequently occurs when upper glycolysis outpaces lower glycolysis (*van Heerden et al., 2014*). Under these conditions, inorganic phosphate becomes a limiting factor for growth as the 'investment' steps in upper glycolysis deplete the cells of ATP and phosphate to form sugar phosphates (*Teusink et al., 1998*; *van Heerden et al., 2014*). Indeed, *S. uvarum gal80Δ gal80bΔ* strains had one-fifth of the ATP as wild-type prior to the TGA phase (*Figure 5—source data 2*) and had significantly up-regulated (25-fold) expression of *PHO84*, which encodes a high-affinity phosphate transporter (*Figure 5—source data 1*).

*S. cerevisiae* combats metabolic overload in upper glycolysis by using two main pathways to restore phosphate pools. The trehalose cycle temporarily reroutes upper glycolysis to store sugars as trehalose (*van Heerden et al., 2014*), while glycerol biosynthesis offers an early exit from glycolysis (*Luyten et al., 1995*). Disrupting the *S. cerevisiae* trehalose cycle leads to the accumulation of fructose-1,6-biphosphate, decreased ATP levels, and ultimately growth arrest due to a metabolically unbalanced state (*van Heerden et al., 2014*; *Gibney et al., 2015*), metabolic changes similar to the *S. uvarum* TGA phenotype. Strikingly, both pathways experienced dramatic bottlenecks in *S. uvarum gal80Δ gal80bΔ* strains before and during the TGA phase. Specifically, *S. uvarum gal80Δ gal80bΔ* cells accumulated 79- to 231-fold more trehalose-6-phosphate before and during the TGA phase, while they accumulated 225-fold more glycerol-3-phosphate before the TGA phase, the latter of which lessened to some extent during the TGA phase (3- to 16-fold) (*Figure 5* and *Figure 5—source data 2*). These data are consistent with the hypothesis that the trehalose cycle and the glycerol biosynthesis pathway are unable to handle the metabolic overload when galactose is catabolized too rapidly in *S. uvarum* strains lacking the *GAL* network repression system.

The metabolic effects of the TGA phenotype also reverberated downstream, leading to the transcriptional down-regulation of the lower part of the TCA cycle and the electron transport chain (*Figure 5*). Reduced respiratory activity has been shown to increase the formation of reactive oxygen species (ROS) (*Barros et al., 2004*), and the co-repressor double mutant had strong signatures of mitochondrial dysfunction. GO terms related to mitochondrial structural components, mitochondrial translation, and respiration were among the most strongly down-regulated (*Supplementary file 1*). Indeed, we observed significantly higher accumulation of ROS in *S. uvarum gal80Δ gal80bΔ* during

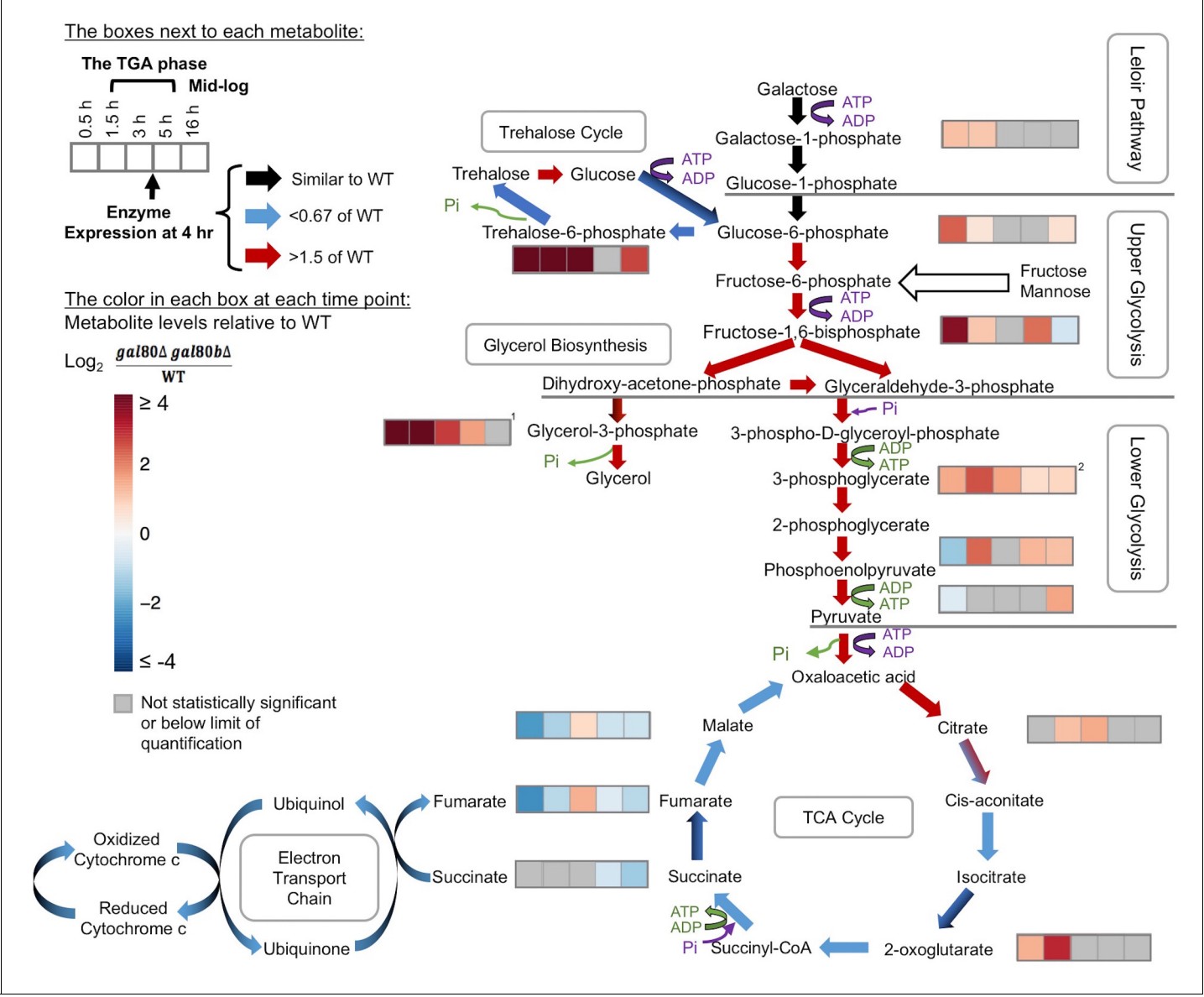

**Figure 5.** Overly rapid galactose catabolism leads to metabolic overload and bottlenecks. The graph shows the metabolite levels and transcript expression for the Leloir pathway, glycolysis, trehalose cycle, glycerol biosynthesis, TCA cycle, and electron transport chain. Purple steps cost ATP or inorganic phosphate ($P_i$), while green steps generate ATP or $P_i$. Strains were cultured in SC + 2% galactose. The arrows are color-coded to represent the RNA-Seq gene expression differences of *S. uvarum gal80Δ gal80bΔ* relative to wild-type at 4 hr (red, up in the mutant; blue, down in the mutant; black, similar to wild-type; mixed colors (e.g. black and blue) indicate that the expression of genes involved in this step differs). The boxes next to each metabolite represent the $\log_2$ of the metabolite concentration differences relative to wild-type over time (0.5, 1.5, 3, 5, and 16 hr, respectively). The statistical significance for metabolite levels was assessed using Student's t-tests (*n* = 3, p<0.05 with gray reported as not significant). The 1.5 hr to 5 hr time points correspond to the TGA phase, whereas the 16 hr time point corresponds to mid-log phase after recovery from the TGA phase. [1], the sum of the metabolite concentrations of glycerol-3-phosphate and glycerol-2-phosphate, the latter of which is not known to be a major metabolite in *Saccharomyces*; [2], the sum of the metabolite levels of 3-phosphoglycerate and 2-phosphoglycerate.

The following source data and figure supplement are available for figure 5:

**Source data 1.** Differential gene expression between *S. uvarum gal80Δ gal80bΔ* and wild-type during the TGA phase and at mid-log phase.

**Source data 2.** Mass spectrometry metabolomic results comparing *S. uvarum gal80Δ gal80bΔ* to wild-type during the TGA phase and mid-log phase.

**Figure supplement 1.** Galactose-1-phosphate accumulation of *S. cerevisiae gal7Δ* and *gal10Δ*.

the TGA phase by using the general ROS indicator dichlorodihydrofluorescein diacetate ($H_2$DCF-DA) (*Figure 4C*). We conclude that disconnecting *S. uvarum* galactose metabolism from the negative feedback loops normally provided by the co-repressors Gal80 and Gal80b likely allows galactose to enter the Leloir pathway and glycolysis too rapidly, leading to metabolic defects far beyond the mild accumulation of galactose-1-phosphate and deep into central metabolism.

## Specific sugars can exacerbate metabolic overload

To determine whether the TGA phenotype reflected a more general metabolic constraint imposed by the interplay between glycolysis and interacting metabolic pathways, we grew *S. uvarum gal80Δ gal80bΔ* in mixtures of galactose with fructose, mannose, or glucose. Fructose, mannose, and glucose are all primarily catabolized through glycolysis, but only glucose generates glycolytic intermediates that are upstream of the trehalose cycle (*Figure 5*). Thus, fructose and mannose are expected to contribute directly to metabolic overload with minimal offsetting effects from the trehalose cycle. If the interaction between glycolytic load and the trehalose cycle were important to the TGA phenotype, growing the double mutant in mixtures of galactose with fructose or mannose would exacerbate the growth arrest. In contrast, if the TGA phenotype were caused by galactose-specific metabolism, the addition of these more preferred sugars would have no effect, or perhaps mitigate the TGA phenotype. Consistent with the TGA phenotype being caused by a general overloading of upper glycolysis, both fructose and mannose strongly exacerbated the TGA phenotype in *S. uvarum gal80Δ gal80bΔ*, while glucose partially rescued the TGA phenotype (*Figure 6A*). Importantly, mixing fructose or mannose with galactose had much stronger defects than the identical amounts of galactose alone (*Figure 4B* and *Figure 6A*). Co-culturing wild-type *S. uvarum* strains in galactose with these sugars was not inherently toxic (*Figure 6A*), so the presence of the co-repressors allows cells to cope with this challenge. Growing *S. uvarum gal80Δ gal80bΔ* in fructose, mannose, or glucose alone also did not cause growth defects (*Figure 4—figure supplement 5*). Moreover, deleting *GAL1* completely rescued the TGA phenotype in the co-repressor double mutant (*Figure 6B*), while mixtures of mannose and galactose dramatically increased the levels of ROS in *S. uvarum gal80Δ gal80bΔ* (*Figure 6C*), implying that the phenotypic enhancement caused by this sugar mixture acts through the same mechanism observed in galactose alone. Collectively, these results suggest that overly rapid catabolism of sugars can lead to general metabolic and growth defects when the appropriate futile metabolic cycles and negative feedback regulatory loops are not able to slow down catabolism.

## The less active *S. cerevisiae GAL* network is less susceptible to metabolic overload when derepressed

We next considered whether the differences between the *GAL* networks of *S. cerevisiae* and *S. uvarum* might explain why a similar phenotype had not been reported for *S. cerevisiae* co-repressor mutants. Recent work has convincingly shown that the *S. uvarum GAL* network is more transcriptionally active than the *S. cerevisiae GAL* network, especially in non-inducing and mixed sugar conditions (*Caudy et al., 2013*; *Roop et al., 2016*). Thus, we wondered whether *S. cerevisiae* and *S. uvarum* galactose catabolism might be under qualitatively similar constraints, even as the more poised and active state of the *S. uvarum GAL* network might render it more vulnerable to metabolic overload. First, we examined *S. cerevisiae gal80* null mutants more closely and found a similar but less-pronounced early rapid increase in optical density, followed by a brief but reproducible TGA phenotype (*Figure 4A*, inset). This observation was missed by earlier studies, which were focused on later time points, because *S. cerevisiae gal80* null mutants eventually grow much faster on galactose (*Torchia et al., 1984*; *Segrè et al., 2006*; *Hittinger et al., 2010*). To test whether the weak TGA phenotype seen in *S. cerevisiae* was due to mechanistically similar metabolic constraints, we sought to exacerbate the phenotype of a *S. cerevisiae gal80Δ* strain in a mixture of mannose and galactose. Indeed, the co-repressor mutant produced significantly more ROS than wild-type under these conditions (*Figure 7A*) and grew slightly more slowly (*Figure 7B*).

Given the interspecific functional differences described above for *GAL1* (*Figure 2C*) and its role as the gatekeeper of the Leloir pathway, we hypothesized that the varied strengths of the TGA phenotype might be due to genetic differences in the *GAL1* locus. Thus, we precisely replaced the *S. uvarum GAL1* promoter or the *GAL1* coding sequence with their *S. cerevisiae* counterparts in *S.*

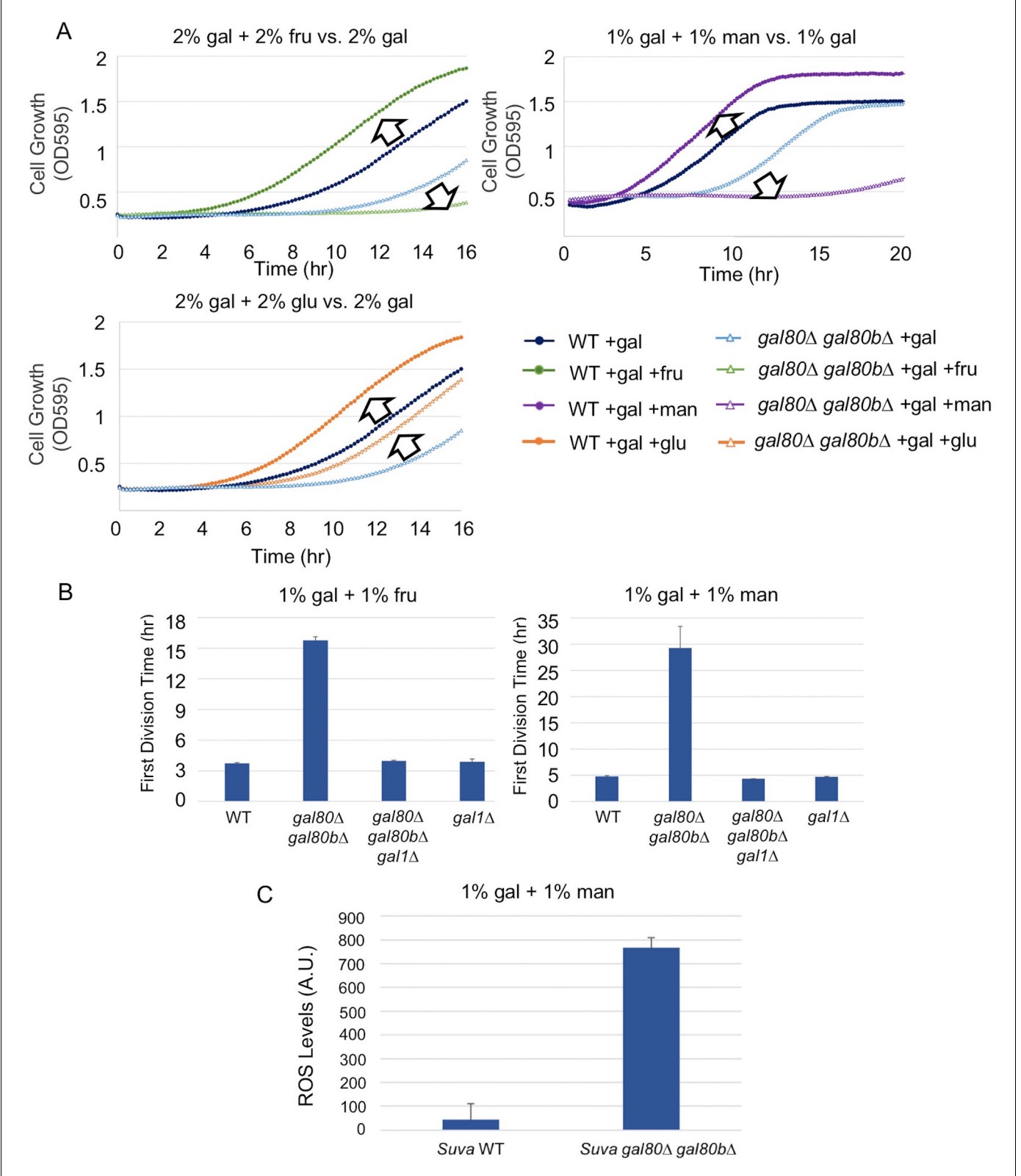

**Figure 6.** The addition of sugars downstream of the trehalose cycle exacerbated metabolic overload. (**A**) Fructose and mannose exacerbated the TGA phenotype in the *S. uvarum gal80Δ gal80bΔ* background, whereas glucose partially rescued the TGA phenotype. (**B**) The *S. uvarum* TGA phenotype in galactose and fructose or mannose can be rescued by the deletion of *GAL1*. The average times to first doubling are shown for three biological replicates. The error bars represent standard deviations. *S. uvarum gal80Δ gal80bΔ gal1Δ* was significantly different than *S. uvarum gal80Δ gal80bΔ* in

*Figure 6 continued on next page*

*Figure 6 continued*

both SC + 1% galactose +1% fructose (p=4.8e-3, *n* = 6, Wilcoxon rank sum test) and SC + 1% galactose + 1% mannose (p=2.9e-3, *n* = 6, Wilcoxon rank sum test). *S. uvarum gal80Δ gal80bΔ gal1Δ* was not significantly different from *S. uvarum gal1Δ* in SC + 1% galactose +1% fructose (p=0.43, *n* = 6, Wilcoxon rank sum test) but was marginally different from *S. uvarum gal1Δ* in SC + 1% galactose +1% mannose (p=0.03, *n* = 6, Wilcoxon rank sum test). (C) Elevated accumulation of ROS in *S. uvarum gal80Δ gal80bΔ* in SC + 1% galactose +1% mannose. *S. uvarum gal80Δ gal80bΔ* had significantly higher ROS levels than the wild-type (p=8.6e-6, *n* = 11, Wilcoxon rank sum test). ROS levels are reported as relative fluorescence levels.

*uvarum gal80Δ gal80bΔ*. The *S. cerevisiae GAL1* promoter rescued the TGA phenotype to some extent, but the *GAL1* coding sequence swap was able to rescue the TGA phenotype to an even greater extent (*Figure 7C*). To confirm that *ScerGAL1* was less active than *SuvaGAL1* and not less toxic for other reasons, we examined the same precise allele replacements in an otherwise wild-type *S. uvarum* strain (i.e. containing functional copies of both co-repressors), as well as a precise reciprocal swap in *S. cerevisiae* replacing the *ScerGAL1* promoter with the *SuvaGAL1* promoter. Swapping the *ScerGAL1* promoter and coding sequence into *S. uvarum* both led to lower growth rates in galactose, while swapping the *SuvaGAL1* promoter into *S. cerevisiae* led to faster growth (*Figure 7D*). We conclude that the *S. uvarum GAL1* promoter and coding sequences both encode higher activity than their *S. cerevisiae* counterparts. Thus, differences in their *GAL* network activities at least partly explain the relative strengths of their TGA phenotypes and the constraints placed on their galactose metabolisms.

## Discussion

### Biodiversity offers a panoramic window to molecular biology

The deep conservation of metabolism and many molecular processes contrasts sharply with the rapid turnover in the regulatory networks that sculpt organismal and phenotypic diversity. Here we have shown how numerous genetic differences between the *S. cerevisiae* and *S. uvarum GAL* networks, especially in the functions of paralogous regulatory genes, contribute to a more poised and active state in *S. uvarum* that is coupled to more robust repression system. When genes encoding the co-repressors were deleted, *S. uvarum* displayed a strong and unexpected growth arrest in galactose, likely due to metabolic overload. Even though *S. cerevisiae* produced qualitatively similar results, decades of previous research on this iconic metabolic and regulatory network overlooked their relatively mild presentation. Just as exaggerated manifestations facilitated the discoveries of transposons in maize, RNAi in *Caenorhabditis elegans*, and telomeres in *Tretrahymena* (*Blackburn et al., 2006*), the striking phenotype observed in the non-traditional model organism *S. uvarum* allowed us to more fully characterize the defect caused by overly rapid galactose catabolism, while demonstrating metabolic constraints conserved across sugars and organisms.

### The non-equivalence of sugars in contributing to metabolic overload

In contrast to glucose, fructose and mannose each had strikingly deleterious effects on cells that were already consuming galactose too rapidly. In *Saccharomyces*, these differences can be explained both by their effects on signaling pathways and by their entry points into glycolysis. Several glucose signaling pathways directly repress *GAL* gene transcription (*Johnston et al., 1994*) and increase the degradation rate of Gal2 protein (*Horak and Wolf, 2001*), both of which would serve to reduce glycolytic load. In *S. cerevisiae*, fructose and mannose do not trigger glucose repression as strongly as glucose (*Dynesen et al., 1998*; *Meijer et al., 1998*). Perhaps as importantly, fructose and mannose bypass the trehalose cycle, a futile cycle recently shown to detour more than a quarter of early-stage glycolytic flux to prevent an unbalanced metabolic state and growth arrest (*van Heerden et al., 2014*). The challenges of the catabolism of sugars other than glucose may be widespread. For example, in humans, bypassing glucose-responsive regulatory mechanisms with fructose has been associated with diabetes (*Lê et al., 2009*; *Kolderup and Svihus, 2015*) and cancer (*Port et al., 2012*; *Jiang et al., 2016*).

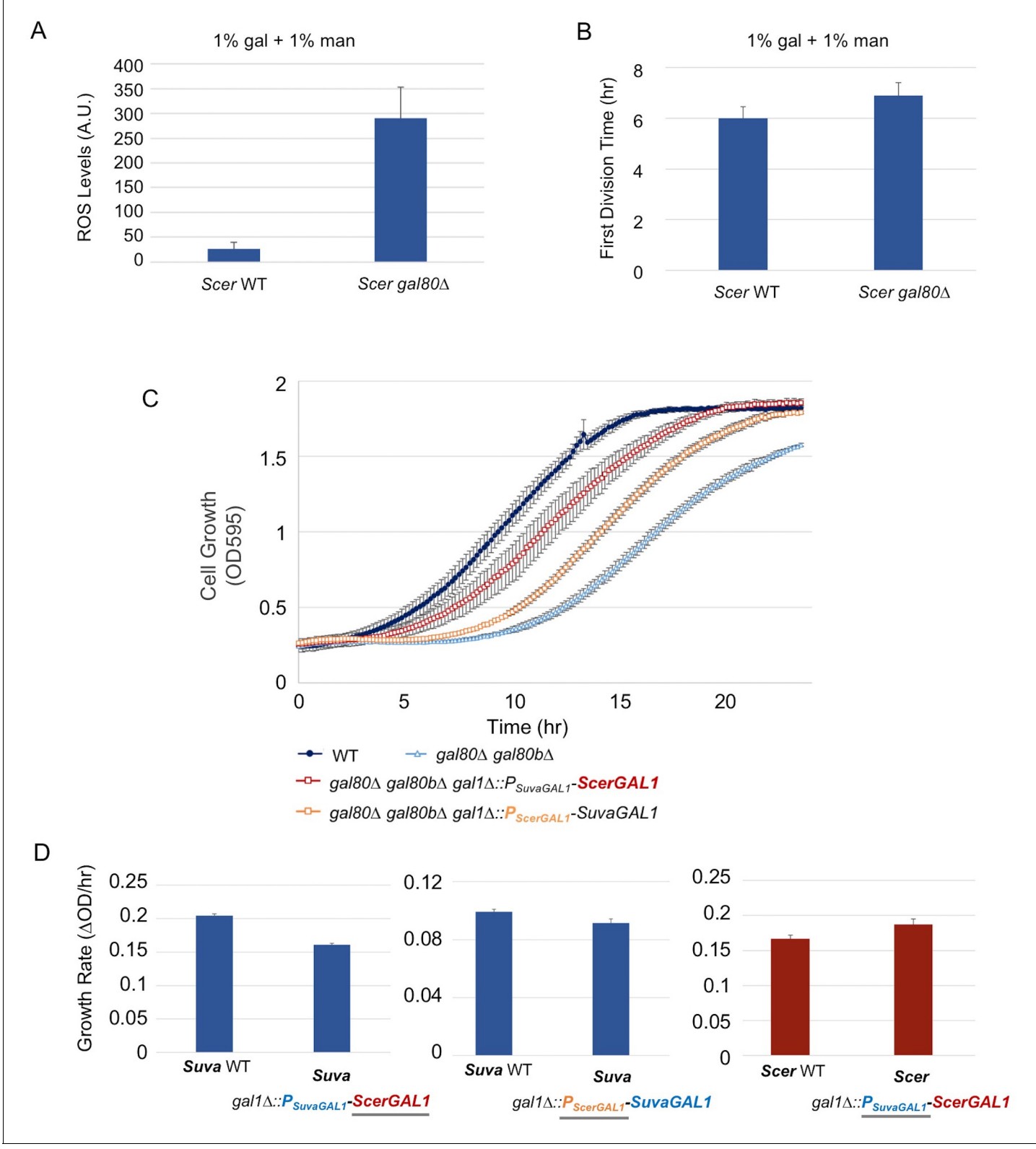

**Figure 7.** The less active *S. cerevisiae GAL1* gene is partially responsible for a subtle temporary growth arrest. (**A**) Elevated accumulation of ROS in *S. cerevisiae gal80Δ* in SC + 1% galactose +1% mannose. *S. cerevisiae gal80Δ* had significantly higher ROS than wild-type (p=2.3e-6, *n* = 12, Wilcoxon rank sum test). ROS levels are reported as relative fluorescence levels. (**B**) *S. cerevisiae gal80Δ* grew more slowly when galactose was mixed with mannose. The average of three biological replicates from a representative experiment is shown, and the error bars represent standard deviations

*Figure 7 continued on next page*

*Figure 7 continued*

(p=0.028, *n* = 6, Wilcoxon rank sum test). (**C**) Both the *ScerGAL1* coding sequence and promoter are able to partially rescue the TGA phenotype. The error bars show the standard deviation of three biological replicates. (**D**) Both the *ScerGAL1* coding sequence and promoter reduced the growth rate of an otherwise wild-type strain of *S. uvarum* in SC + 2% galactose, while the reciprocal swap of the *GAL1* promoter in *S. cerevisiae* increased its growth rate. Wilcoxon rank sum tests comparing the specific growth rates of each subpanel were all significant: (1) p=2.3e-6 and *n* = 12 for *S. uvarum gal1Δ::*$P_{SuvaGAL1}$-*ScerGAL1* versus *S. uvarum* wild-type, (2) p=2.5e-4 and *n* = 9 for *S. uvarum gal1Δ::*$P_{ScerGAL1}$-*SuvaGAL1* versus *S. uvarum* wild-type, and (3) p=8.8e-3 and *n* = 9 for *S. cerevisiae gal1Δ::*$P_{SuvaGAL1}$-*ScerGAL1* versus *S. cerevisiae* wild-type.

## Network architectures reflect metabolic constraints

The intrinsic constraints imposed by galactose metabolism may have led to the evolution of regulatory mechanisms that protect against the risks of metabolic overload. Many of the differences between the *S. uvarum* and *S. cerevisiae GAL* networks can be explained as offering alternative protective strategies, while affording varied catabolic capabilities. For instance, the direct regulation of the *PGM1* gene by Gal4 would enhance the connection between the Leloir pathway and glycolysis in *S. uvarum* relative to *S. cerevisiae* (*Fu et al., 2000*; *Ostergaard et al., 2000*; *Garcia Sanchez et al., 2010*). *S. uvarum PGM1* is highly induced by galactose (*Figure 4—figure supplement 6*), but this likely ancestral regulatory connection was lost in the *S. cerevisiae-S. kudriavzevii* clade (*Figure 4—figure supplement 6*). Nearly all of the known differences between the *S. cerevisiae* and *S. uvarum GAL* networks make *S. uvarum* more active, including (1) apparent regulation of *PGM1* by Gal4; (2) the presence of genes encoding two galactose transporters (*Figure 1*); (3) the galactokinase activity of SuvaGal3 (*Figure 2A*); (4) the higher activity of both the *GAL1* coding and cis-regulatory sequences (*Figure 7D*); and (5) higher background gene expression across the network (*Caudy et al., 2013*; *Roop et al., 2016*). Indeed, the possession of a gene encoding a second co-repressor appears to be one of the few features of the *S. uvarum GAL* network that would serve to counteract its higher activity. Thus, the dramatic up-regulation of *GAL80B* during induction may offer a robust negative feedback loop that helps prevent over-induction and metabolic overload. The retention of *GAL80B* may have allowed *S. uvarum* to maintain a more active *GAL* network, while the *S. cerevisiae GAL* network evolved lower activity.

Comparison of yeast genomes beyond the *Saccharomyces* suggests that galactose metabolism may impose similar constraints across the yeast phylogeny. The genes encoding the Leloir enzymes occur in one of the few broadly conserved yeast gene clusters (*Wong and Wolfe, 2005*; *Slot and Rokas, 2010*; *Wolfe et al., 2015*; *Riley et al., 2016*), which has been suggested could promote enzyme co-regulation to prevent the accumulation of toxic intermediates (*Price et al., 2005*; *Lang and Botstein, 2011*) or ensure that only complete networks are co-inherited (*Lawrence and Roth, 1996*; *Hittinger et al., 2010*). In addition to *S. uvarum*, many yeast species that underwent the WGD retain *GAL80B* (*Hittinger et al., 2004*). Perhaps due to these intrinsic metabolic challenges and the limited benefits of maintaining a dedicated *GAL* network, the ability to consume galactose has been lost many times across diverse yeast lineages (*Hittinger et al., 2004*, *2015*; *Slot and Rokas, 2010*; *Wolfe et al., 2015*; *Riley et al., 2016*).

## Ongoing functional diversification of paralogs and their gene networks

In contrast to more commonly studied processes of the rapid neofunctionalization and subfunctionalization of paralogs (*Moore and Purugganan, 2003*; *Innan and Kondrashov, 2010*), we have shown how duplicate *GAL* genes continued to diverge functionally in ways that dramatically influenced the metabolic and regulatory states of extant *Saccharomyces* species. Based on the redundancy observed between *GAL1* and *GAL3* and between *GAL80* and *GAL80B* in *S. uvarum*, we infer that the functions of these two paralog pairs overlapped more at the origin of the genus *Saccharomyces* than in *S. cerevisiae* (*Figure 8*). After the *S. uvarum-S. eubayanus* clade diverged from the *S. arbori-cola-S. cerevisiae* clade, these genes met distinct fates in different lineages (*Figure 8*). *GAL80B* was lost in the *S. cerevisiae-S. arboricola* clade, while it was retained in *S. uvarum* and *S. eubayanus* (*Hittinger et al., 2010*, *2004*; *Scannell et al., 2011*; *Caudy et al., 2013*; *Hittinger, 2013*; *Liti et al., 2013*; *Baker et al., 2015*). The fates of *GAL1* and *GAL3* were still more varied. *GAL3* was lost in a European population of *S. kudriavzevii*, resulting in an induction defect, while the entire *GAL* network was lost in an East Asian population of this species that cannot consume galactose

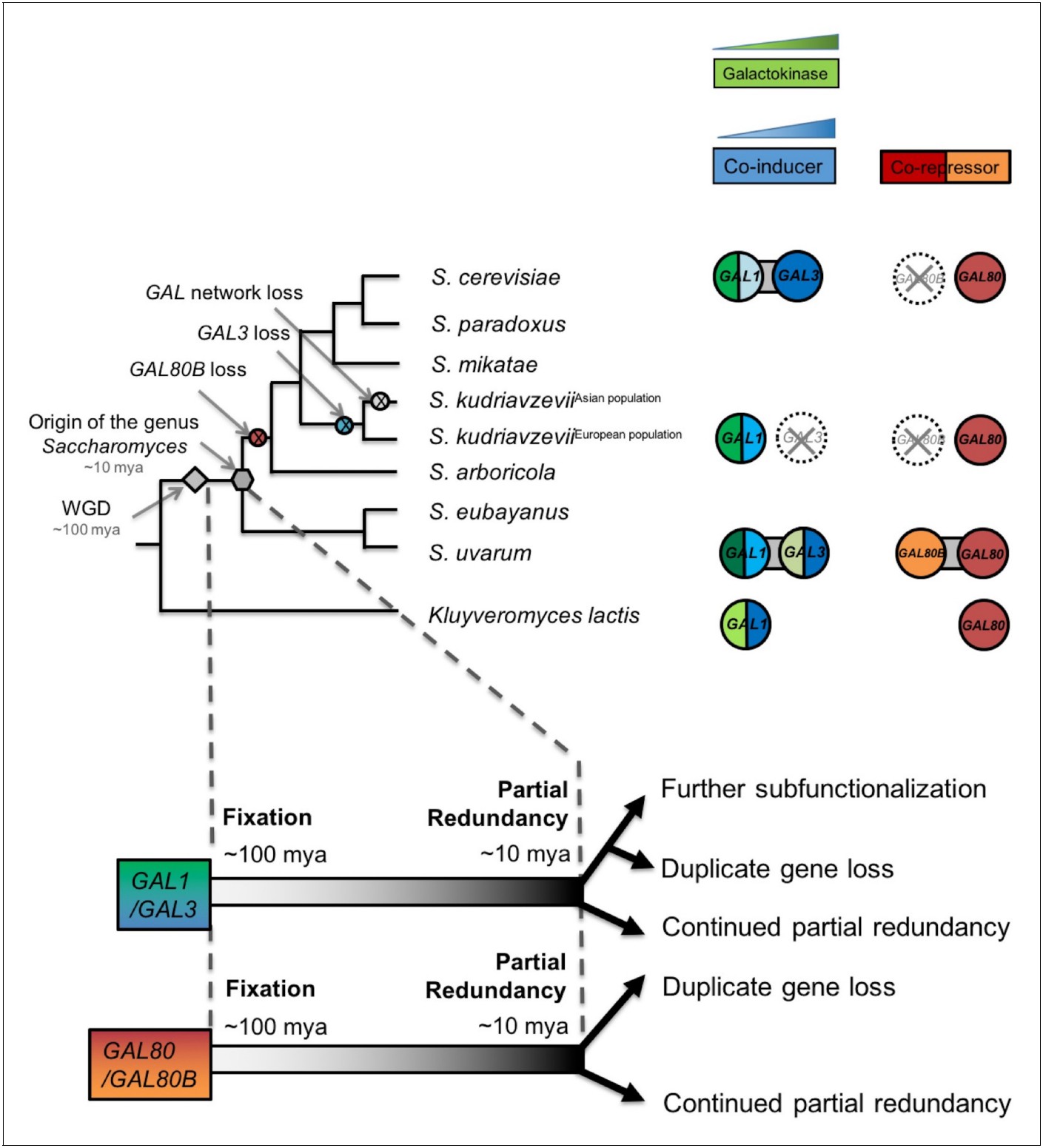

**Figure 8.** Ongoing diversification of the functions of the *GAL1-GAL3* and *GAL80-GAL80B* duplicate gene pairs in *Saccharomyces*. Important evolutionary events are shown on the cladogram. WGD, the whole genome duplication that created the two pairs of paralogs. The inferred duplicate divergence fates are shown at the bottom of the tree. The inferred timeline is depicted by the dashed line. Roughly ~100 million years ago, these two pairs of duplicate genes were fixed in the ancestral genome following a WGD event. Considerable partial redundancy was maintained in the lineage leading to the origin of the genus *Saccharomyces*. In the last ~10 million years, the fates of the duplicate genes have functionally diverged along different evolutionary trajectories. The bifunctionality of the *GAL1/GAL3* genes is represented by green for the enzymatic function and blue for the co-

*Figure 8 continued on next page*

Kuang *et al*. eLife 2016;5:e19027. DOI: 10.7554/eLife.19027

*Figure 8 continued*

induction function. The color shading represents approximate functionality for experimentally characterized genes: a darker color indicates a stronger function, whereas a lighter color indicates a weaker function. The dashed circle with a cross indicates the loss of the indicated gene. Note that the *S. kudriavzevii* Asian population lost its entire *GAL* network, while the European population retained most of its *GAL* network but lost *GAL80B* and *GAL3*. The additional co-repressor in *S. uvarum* may minimize the risk of metabolic overload due to an otherwise highly active *GAL* network.

(*Hittinger et al., 2010*). *GAL1* and *GAL3* were nearly completely subfunctionalized in *S. cerevisiae* (*Hittinger and Carroll, 2007*), but we have shown here that they maintain considerable redundancy in *S. uvarum*.

For both paralog pairs, the ongoing functional diversification has been asymmetric. Deleting *GAL80B* and *GAL3* produced less striking phenotypes than deletion of their paralogs in *S. uvarum*, and some lineages have experienced inactivation or loss of these genes naturally. In the lineage leading to *S. cerevisiae*, Gal3 completely lost enzymatic activity, while a decrease in the promoter activity of its paralog *GAL1* reduced, but did not eliminate, its ability to induce the network rapidly. Other *GAL* genes also experienced an adaptive decrease in promoter activities in the lineage leading to *S. cerevisiae* (*Roop et al., 2016*), which may have been enabled or necessitated by the loss of the secondary co-repressor encoded by *GAL80B*. Remarkably, the disparate resolutions of the functions of these paralogs did not happen soon after the WGD that created them. Instead, the diversification described here occurred within the last 10 million years of a 100 million year history, demonstrating that the echoes of duplication events continue to resonate through gene networks much longer than is generally appreciated (*Gordon et al., 2009*; *Conant et al., 2014*).

The ongoing functional diversification of ancient paralogs likely has an even greater impact on the evolution of plants and vertebrates, where nearly all extant species are the products of multiple rounds of WGD, and differential paralog retention is widespread (*Amores et al., 1998*, *2004*; *Aury et al., 2006*; *Blomme et al., 2006*; *Gómez-Valero et al., 2007*; *Scannell et al., 2007*; *De Smet et al., 2013*; *McGrath et al., 2014*). Molecular and genetic dissection is much more challenging in these systems, but there are hints that the diversification of ancient paralogs continues to have functional consequences for the evolution of metabolism (*Steinke et al., 2006*; *Conant et al., 2014*) and development (*Kassahn et al., 2009*; *Cortesi et al., 2015*). Paralog diversification is often asymmetric as one paralog acquires a more specialized or auxiliary role (*Force et al., 1999*; *Moore and Purugganan, 2003*; *Hittinger and Carroll, 2007*; *Des Marais and Rausher, 2008*; *Innan and Kondrashov, 2010*; *Conant et al., 2014*). Even if this specialization is conditionally adaptive, the auxiliary paralog can become more susceptible to gene loss when conditions change. Paralog loss ends the saga of duplicate gene diversification, possibly forcing partially redundant functions back onto the remaining paralog, relieving paralog interference (*Baker et al., 2013*), or leading to compensatory changes elsewhere in the network. Perhaps more interestingly, paralog loss eliminates redundancy and limits the long-term potential for adaptation. The ongoing evolutionary processes affecting the *GAL* paralogs show how gene duplication facilitates phenotypic change and network diversification in ways that continue to reverberate.

## Materials and methods

### Strain construction

To construct *GAL* gene knockouts, we used MX cassettes (*hphMX*, *natMX*, or *kanMX*) (*Wach et al., 1994*; *Goldstein and McCusker, 1999*) to precisely replace the coding sequence from start codon to stop codon. Transformations were based on the standard lithium acetate/PEG method optimized for *S. uvarum* (room temperature incubation, followed by a 37°C heat shock) (*Gietz et al., 1995*; *Caudy et al., 2013*). To perform allele swaps, the coding sequence or promoter was first replaced by a selectable and counter-selectable *TK-hphMX* cassette, which does not require the prior introduction of an auxotrophy (*Alexander et al., 2014*). The coding sequence or promoter of the desired replacement sequence was amplified by PCR primers with overhangs homologous to the targeted genomic flanking region. In some cases, extended homology (100–300 bp) was then introduced through PCR sewing. For each *GAL1* promoter swap, we swapped the entire upstream intergenic

region. Note that the *S. cerevisiae* and *S. uvarum GAL1* promoters are both divergent promoters that also regulate *GAL10* and may also impact a lncRNA previously described in *S. cerevisiae* (*Cloutier et al., 2016*). Successful replacement strains were isolated by selecting for the loss of thymidine kinase activity by resistance to 5-fluorodeoxyuridine (FUdR), as well as the loss of resistance to hygromycin by replica plating (*Alexander et al., 2014*). GFP reporters were constructed in three parts: the *hphMX* cassette was placed upstream as the selection marker, the *S. uvarum GAL1* promoter was used to drive the expression of the reporter, and the reporter was a *yEGFP* (yeast Enhanced Green Fluorescence Protein) construct with a *S. cerevisiae CYC1* terminator that was amplified from FM1282 (*Hittinger and Carroll, 2007*; *Hittinger et al., 2010*). GFP reporters were introduced to replace *S. uvarum gto1*, an inactive pseudogene (chr7: 767,328–766,478) orthologous to *S. cerevisiae GTO1* (*Scannell et al., 2011*). The modified loci of all transformants were verified by Sanger sequencing. *S. cerevisiae* is NCBITaxon:4932, *S. uvarum* is NCBITaxon:230603, and the strains used in this study are listed in *Supplementary file 2*.

## Media and growth assays

Strains were first streaked on YPD (10 g/L yeast extract, 20 g/L peptone, 20 g/L glucose, 18 g/L agar) plates from frozen glycerol stocks. Next, a single colony of each strain was cultured in synthetic complete (SC) medium plus 0.2% glucose (1.72 g/L yeast nitrogen base without amino acids, 5 g/L ammonium sulfate, 2 g/L complete dropout mix, 2 g/L glucose) for 2–3 days, a condition that does not induce and only minimally represses the *GAL* network. There were at least two biological replicates for each genotype, generally from independent transformants. These pre-cultures were washed with water and inoculated into the desired growth media in a 96 well plate. No explicit power analyses were performed to determine sample sizes or the number of replicates. Instead, each experiment was independently performed at least twice on separate days; details can be found in each legend. Biological replicates were defined as independent isogenic colonies on agar plates, which were used for subsequent precultures and growth assays; technical replicates were defined as independent growth assays from the same preculture. The absorbance of each well was read by an unshaken BMG FLUOstar Omega plate reader every 10 min at 595 nm. The number of cell divisions for each time point was calculated as $\log_2[(\mathrm{OD}_{strain} - \mathrm{OD}_{media})/(\mathrm{OD}_{start} - \mathrm{OD}_{media})]$, an equation that normalized each optical density time point to its starting optical density and the optical density of the medium. The times to first doubling were calculated as the times for the optical densities to double from their normalized starting points. Specific growth rates were calculated using the Growth Curve Analysis Tool (GCAT) (*Bukhman et al., 2015*). Replicates that failed to grow as precultures or during growth assays were considered as outliers and were excluded from subsequent analyses; no other data were excluded. For *S. cerevisiae* and *S. uvarum gal1* mutant growth assays (*Figure 2—figure supplement 1*), strains were pre-cultured in SC plus 0.67% fructose for 2 days and inoculated at a 1:1000 ratio into supplemented minimal medium (1.72 g/L yeast nitrogen base without amino acids, 5 g/L ammonium sulfate, 85.6 mg/L uracil, 85.6 mg/L lysine, 20 g/L galactose) plus 2% galactose or no carbon source. The growth properties of these strains were determined by subtracting the optical densities of cultures in media without a carbon source from media with galactose; differences less than 0.05 were considered as 'no growth.' In each case, *S. cerevisiae* strains were cultured at 30°C, while *S. uvarum* strains were cultured at 24°C, except when they were cultured in the same 96 well plate. In these cases (*Figure 2C*, *Figure 2B* and *Figure 4A*), strains were grown at 26°C, and the results were summarized in one graph.

## Flow cytometry

The pre-culture and growth conditions were identical to those described above for the 96-well growth assays. At the indicated time points, 1–30 µL cultures were transferred from the 96-well plate to fresh medium of the same type in a new 96-well plate to obtain a concentration of 200–500 cells/µl for flow cytometry. There were at least three biological replicates for each genotype. The flow cytometry was conducted using a Guava EasyCyte Plus flow cytometer. Each experiment was independently conducted at least twice on separate days. The data were extracted from FCS 2.0 formatted files using FlowCore (*Hahne et al., 2009*) (RRID:SCR_002205). The fluorescence levels were normalized by forward scatter to control for cell size. For each genotype, histograms of normalized

fluorescence levels of 6000 cells were smoothed by Kernel density estimation and plotted using the R statistical package.

## RNA sequencing

Strains were pre-cultured in SC plus 0.2% glucose for 2 days and inoculated into SC plus 2% galactose, 2% glucose, or 5% glycerol. Samples were harvested at the indicated time points and frozen using a dry ice/ethanol bath. Total RNA was extracted using the standard acidic phenol protocol (*Hittinger and Carroll, 2007*), and residual DNA was removed through DNase I treatment. Poly-A enrichment was performed with the NEBNext Poly(A) mRNA Magnetic Isolation Module (NEB #E7490, in the experiment to examine *S. uvarum GAL* network membership) or with the NEB Magnetic mRNA Isolation kit (NEB #S1550, in the experiment sampled during the TGA phase and at mid-log phase in galactose). Illumina libraries were constructed using the NEB Ultra Directional RNA Library Prep Kit for Illumina (NEB #E7420) and sequenced using the Illumina HiSeq 2500 platform. Reads were mapped onto the *S. uvarum* reference genome (CBS7001) (*Scannell et al., 2011*) using Bowtie version 2.2.2 with local read alignment and otherwise default settings (*Langmead et al., 2009*). Read counts were quantified by HTSeq version 0.6.0 (*Anders et al., 2015*) (RRID:SCR_005514). Differential expression was determined using EBseq version 1.1.5 with a false discovery rate (FDR) of 0.05 (*Leng et al., 2013*) (RRID:SCR_003526). Analysis with edgeR (RRID:SCR_012802) using the default settings was performed in parallel to examine known *S. cerevisiae* Gal4 targets that were not scored as differentially expressed in *S. uvarum* (*Robinson et al., 2010*). Differentially expressed genes were further analyzed by GO term analysis (*Ashburner et al., 2000*; *Cherry et al., 2012*) (Generic GO Term Mapper, RRID:SCR_005806; SGD Gene Ontology Slim Mapper, RRID:SCR_005784). The RNA-Seq data are available at NCBI's SRA under accession number SRP077015.

## Reactive oxygen species (ROS) measurements

The pre-culture conditions were identical to those described above for the growth assays. The ROS measurement protocol was adapted from a previous study (*Dudgeon et al., 2008*). A 10 mM stock of $H_2DCFDA$ (2′,7′-dichlorodihydrofluorescein diacetate) was freshly prepared in ethanol before each use. Cells were washed once and inoculated into the stated growth medium with 10 μM $H_2DCFDA$. Cultures were harvested at the indicated time points. Fluorescence levels and optical densities were measured using a BMG FLUOstar Omega plate reader, which can read both fluorescence and optical density. To establish standard curves, a 2-fold serial dilution for each strain was measured. Since the standard curves suggested a linear relationship between fluorescence levels and cell number, fluorescence levels were normalized to optical densities. The *S. uvarum* wild-type strain was inoculated into YPD plus 10 mM $H_2O_2$ and into YPD only as positive and negative controls, respectively. Each experiment was independently conducted at least twice on separate days with at least three biological replicates in each experiment.

## $^{13}$C-labelled yeast metabolome extract preparation

The $^{13}$C yeast metabolome extract (*Bennett et al., 2008*) was prepared by growing Y22-3 (*McIlwain et al., 2016*) aerobically on YNB (-AA) + 1% $^{13}$C glucose. Yeast cultures were inoculated at an OD of 0.05 into $^{13}$C medium. Samples were harvested from each culture by centrifugation and frozen in liquid $N_2$. Frozen pellets were first extracted with 750 μL of 40:40:20 ACN/MeOH/$H_2O$, followed by a second extraction with 500 μL of the same extraction solvent. Extracts were pooled, centrifuged, and the supernatant was collected for later use as an internal standard for absolute metabolite quantification (*Bennett et al., 2008*).

## Liquid chromatography-tandem mass spectrometry (LC-MS/MS) metabolomic analyses

Lyophilized cell culture metabolites were extracted from mutant and wild-type strains with 5 mL ice-cold 7:2:1 MeOH/CHCl$_3$/$H_2O$, and 100 μL of the extract was mixed with 10 μL $^{13}$C-labelled yeast metabolome extract. Three biological replicates were included for the *S. uvarum* strains (*Figure 5*), while two were included for the *S. cerevisiae* strains (*Figure 5—figure supplement 1*). Chromatographic separations based on a previously described method (*van Dam et al., 2002*; *Long et al., 2012*) were carried out on an Agilent 1200 series HPLC comprising a vacuum degasser,

binary pump, heated column compartment, and thermostated autosampler set to maintain 6°C. Mobile phase A (MPA) was 0.5 mM NaOH, and mobile phase B (MPB) was 100 mM NaOH. 20 µL of intracellular extract or calibrant standard mixture was separated on a Dionex IonPac AS11-HC IC column (2.0 mm x 250 mm, 9.0 µm) held at 40°C using a flow rate of 0.35 mL/min. Metabolite elution was achieved by first holding at 5% MPB for 22.5 min to separate isobaric phosphosugar species. MPB was then linearly increased from 5% to 100% over 27.5 min to elute the remaining metabolites. MPB was held at 100% for 7 min for column cleaning followed by an 8 min re-equilibration step at 5% MPB. The LC system was coupled to a Dionex ERS 500 suppressor controlled by a Dionex Reagent-Free Controller (model RFC-10) and an Agilent 6460 A Triple Quadrupole MS. The MS was operated in negative mode, acquiring MRM scans for each metabolite. Quantification based off external standard calibration curves and correction with the [13]C-labelled yeast standard was performed with Agilent MassHunter Quantitative Analysis software (version B.06.00).

## High performance liquid chromatography (HPLC)

The pre-culture conditions were identical to those described above for the growth assays. At indicated time points, 1 mL of cells were centrifuged, and 500 µL supernatant was harvested and frozen at −80°C. HPLC was conducted at the GLBRC Metabolomics Lab using an HPLC-RID system with an Aminex HPX-87H (BioRad, Inc. Hercules, CA) following previously described protocols (*Moore and Johnson, 1967*; *Ehrman and Himmel, 1994*). Instrument control, data collection and analyses were conducted using ChemStation B.04.03 software (Agilent Technologies, Inc., Palo Alto, CA).

## Statistical analysis

All *p*-values, except for the RNA-Seq, metabolomics (two-sided student's t-test), and HPLC analyses (two-sided student's t-test), were calculated using a conservative two-sided nonparametric test. Specifically, we used a Wilcoxon rank sum test that allows the rank data from multiple independent experiments to be pooled to account for day-to-day variation without making assumptions about the distribution of the variance. These tests were performed using Mstat software version 6.1.4 (http://mcardle.oncology.wisc.edu/mstat/).

## Acknowledgements

We thank Mick McGee for performing the HPLC experiments and analyzing the data; Audrey P Gasch for providing access to the Guava flow cytometer; the University of Wisconsin Biotechnology Center DNA Sequencing Facility for providing Illumina and Sanger sequencing facilities and services; Amy A Caudy, Stephen A Johnston, and James E Hopper for helpful discussions; William G Alexander for technical advice on the use of the *TK-hphMX* and *TK-kanMX* cassettes; and Emily Clare Baker and Drew T Doering for critical reading of the manuscript.

## Additional information

### Funding

| Funder | Grant reference number | Author |
| --- | --- | --- |
| National Institutes of Health | R35 GM118110 | Joshua J Coon |
| DOE Great Lakes Bioenergy Research Center | DOE Office of Science BER DE-FC02-07ER64494 | Chris Todd Hittinger |
| National Science Foundation | DEB-1253634 | Chris Todd Hittinger |
| National Institute of Food and Agriculture | Hatch Project 1003258 | Chris Todd Hittinger |
| Pew Charitable Trusts | Pew Scholar in the Biomedical Sciences | Chris Todd Hittinger |
| Alexander von Humboldt-Stiftung | Alfred Toepfer Faculty Fellow | Chris Todd Hittinger |
| National Science Foundation | DEB-1442148 | Chris Todd Hittinger |

The funders had no role in study design, data collection and interpretation, or the decision to submit the work for publication.

## Author contributions
MCK, CTH, Conception and design, Acquisition of data, Analysis and interpretation of data, Drafting or revising the article, Contributed unpublished essential data or reagents; PDH, Acquisition of data, Analysis and interpretation of data, Drafting or revising the article; JDR, JJC, Analysis and interpretation of data, Drafting or revising the article

## Author ORCIDs
Meihua Christina Kuang, http://orcid.org/0000-0003-3206-6525
Chris Todd Hittinger, http://orcid.org/0000-0001-5088-7461

## Additional files

### Supplementary files
• Supplementary file 1. Gene Ontology Term enrichment results of mis-regulated processes in *Suva gal80Δ gal80bΔ* during the TGA phase and mid-log phase.

• Supplementary file 2. List of strains used in this study.

### Major datasets
The following dataset was generated:

| Author(s) | Year | Dataset title | Dataset URL | Database, license, and accessibility information |
|---|---|---|---|---|
| Kuang MC, Hittinger CT | 2016 | RNA-Seq of Saccharomyces uvarum | http://www.ncbi.nlm.nih.gov/sra/SRP077015 | Publicly available at the NCBI Short Read Archive (accession no: SRP077015) |

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
