## [Decision Letter]

Thank you for submitting your article "Ongoing resolution of duplicate gene functions shapes the diversification of a metabolic network" for consideration by *eLife*. Your article has been reviewed by three peer reviewers, including Justin Fay (Reviewer #1), and the evaluation has been overseen by a Reviewing Editor and Aviv Regev as the Senior Editor.

The reviewers have discussed the reviews with one another and the Reviewing Editor has drafted this decision to help you prepare a revised submission.

The reviewers and the Reviewing Editor agree that no additional experiments are required for the main message of the manuscript, although if data is available that addresses some of the questions raised by the reviewers, the authors should include it. The full reviews are included below as they note a number of points where clarification or modification of claims would help the reader and increase the precision with which the results and conclusions is communicated. In particular, please clarify the nature of the strains used (point #2 of reviewer #3) and address the relationship between the metabolic changes and the growth phenotypes (end of second paragraph of reviewer #2: what is the evidence that metabolic overload is causal? Does this inference need to be toned down?)

*Reviewer #1:*

The manuscript describes differences in divergence of two yeast species in their *GAL* network. Their primary finding is that *S. uvarum* has higher activity but also better repression of *GAL* genes compared to *S. cerevisiae*. The repressive component of this difference is shown to be mediated by an ancestral *GAL80b* retained in *S. uvarum* but lost in *S. cerevisiae*. Part of the higher activation is derived from differences in *GAL1*. The work represents a significance advance by dissecting differences in duplicated gene evolution and how they contribute to evolution of the *GAL* network. At a time when we are paying more attention to strain and species-level differences in genetic perturbations, the manuscript exemplifies the kinds of differences we will have to contend with and how we might take advantage of them.

Two sets of experiments make use of precise allele replacements of the *GAL1* promoter. These precise replacements are commendable since they avoid certain issues, however they have the disadvantage of *GAL10* being driven from the same promoter. Can the *GAL1* promoter swaps be attributed to *GAL1* or *GAL10* activity? While the main message of the work does not hinge on the answer, is there data that can address this or can you point this out to the reader. Replacement of the *GAL1* coding region should not be affect by this issue, although there is an interesting lncRNA that covers the *GAL* cluster and is involved in regulation (PMID: 26833086).

The reduced activity of the *GAL* genes in *S. cerevisiae* is striking. However, whenever *S. cerevisiae* is found to have loss of function I always wonder whether this is a defect specific to the lab strain. The lab strain has been found to a have a particularly long lag phase in glucose/galactose mixtures compared to other *S. cerevisiae* strains which also show less *GAL* repression as glucose levels fall (PMID: 25626068). While this concern does not require repeating the experiments in another strain background, it should be addressed in the manuscript, preferably through a single experiment: e.g. *gal80* deletion.

The evidence that *SuvaGAL3* encodes a functional galactokinase needs more support for the strength of the statements made: "*SuvaGAL3* does indeed encode a functional galactokinase". The *Suva gal1* null did not grow on galactose consistent with *Suva GAL3* lacking galactokinase activity. However, when *ScerGAL1* was replaced with *SuvaGAL3* there was growth, indicating sufficient activity for growth because *ScerGAL1*-promoter is strong. Either biochemical evidence is needed that *SuvaGAL3* and *ScerGAL3* differ in their activity or the control is needed where *ScerGAL3* is also overexpressed by the *Scer GAL1* promoter. A last option would be to temper this finding, since it is not essential to the main message.

*Reviewer #2:*

This manuscript describes differences between two budding yeast species in the effects of mutations in the galactose metabolic pathway. The experiments are carefully done and together make up a sizeable body of work. The conclusions are for the most part justified, although I note some exceptions below.

The bulk of the manuscript deals with the discovery of a transient growth arrest in galactose medium in an *S. uvarum* strain in which the two paralogs of the *GAL80* repressor are deleted. This stands in contrast to the growth advantage in *S. cerevisiae* upon deletion of its one *GAL80* copy (Figure 4). Transcriptomic and metabolomic experiments show that in this *S. uvarum* double mutant in galactose, during a short burst of rapid growth followed by arrest, glycolysis is upregulated and the TCA cycle is repressed (Figure 5). Deleting *GAL1* (Figure 6) or replacing the *GAL1* promoter with the *S. cerevisiae* allele (Figure 7) was sufficient to rescue the transient growth arrest phenotype of the *S. uvarum* double *GAL80* mutant, at least in conditions when galactose was mixed with other sugars. Thus the *S. uvarum GAL1* promoter, seemingly by virtue of its high activity (Figure 3), is a driver of the transient growth arrest of the *S. uvarum* double *GAL80* mutant, at least in mixed sugar media. The hyperactivity of the *S. uvarum GAL1* promoter is further underlined by the finding that only it, and not the *S. cerevisiae* allele of the *GAL1* promoter, can compensate for the loss of *GAL3* in an *S. cerevisiae* background during growth in galactose (Figure 2). The authors' model is that the transient growth arrest phenotype of the *S. uvarum* double *GAL80* mutant accrues from overzealous galactose metabolism leading to a broader "metabolic overload." Strictly speaking, any mechanistic role for the suite of expression and metabolite changes seen in this double mutant is not shown.

The authors make the interesting argument that the second paralog of *GAL80, GAL80b*, may have evolved in *S. uvarum* as a necessary brake on its highly active galactose metabolic program. Indeed, they show that *GAL80b* in *S. uvarum* is a stronger repressor of *GAL1* than is *GAL80* (Figure 3). More support for fitness-relevant functions of *GAL80b* could strengthen the manuscript. Since *GAL80* alone is sufficient to prevent the transient growth arrest that otherwise manifests in the *S. uvarum* double *GAL80* mutant in pure galactose (Figure 4—figure supplement 1), any benefit of *GAL80b* wouldn't be in this condition. Rather, the *GAL80b* mutant in *S. uvarum* has a slight growth defect in a mixed sugar environment (Figure 3; it would be good to show the *GAL80* mutant also in this figure). The mechanism of the latter phenotype would be of interest here, particularly in relation to *S. uvarum's* hyperactive galactose metabolic program which is the theme of the work.

A first part of the manuscript, of relatively modest scope, focuses on *GAL3*, which is a regulator in *S. cerevisiae*. Figure 2 shows that *S. cerevisiae* can't grow in galactose when *GAL1* is deleted (an *S. uvarum GAL1* mutant is not reported). This defect can be rescued by the *S. uvarum* allele of *GAL3*, indicating a gain of metabolic function by the latter which the authors infer to be galactokinase activity.

*Reviewer #3:*

I found this paper to be interesting and illuminating, as it captures a dynamic picture or evolution in action. We usually portray a schematic, frozen picture of evolution, focusing on the starting point (e.g. a species that diverged before the yeast WGD) and endpoint (e.g. *S. cerevisiae*). However, as shown here, analysis of multiple species that evolved in parallel shows that evolution is an ongoing process, and that divergence may take different paths, and/or occur to different levels. For example, compared to *S. cerevisiae*, the *S. uvarum GAL1* and *GAL3* genes diverged to lesser extent. However, in *S. uvarum, Gal80* duplicated, and one parallel (*GAL80B*) has diverged further towards galactose triggered expression.

The paper is clearly written, and despite technical complexity and detail-richness, makes an interesting read. The conclusions are in my view well supported, and with adequate controls that use dot be a standard but are now missing in many studies (e.g. reintroduction of the knocked out gene in a different locus to ensure the phenotype is related to this gene missing and not some other genomic perturbation). The study addressed multiple aspects including changes in transcriptome and metabolomics (including the identification of galactose-1-phosphate as the origins of toxicity). While I could imagine (and could not resist suggesting) interesting experiments that may provide further insights, I am impressed by the breadth of the data and there is no need in my view to demand additional experiments.

Comments:

1) *GAL* gene content and sequence differences:

Reading the next section one wonders if the *S. uvarum Gal1/3* the products of the WGD? But then, the authors state: that divergence of "*ScerGAL1* and *ScerGAL3* happened after the divergence of *S. uvarum* and *S. cerevisiae*. I suggest to explicitly discuss *Gal1/3* in this section. Further, although these are not the results of this study, still, a schematic phylogenetic tree(s) for the species and genes addressed here would really help the reader who is not familiar with yeast phylogeny and/or the *Gal* system.

2) "The less active *S. cerevisiae GAL* network is less susceptible to metabolic overload" The data in this section relates to the *GAL80* knockouts of both species. Wouldn't the right comparison be the wild-types?

3) Related to the above: Galactose catabolism in *S. uvarum* can be highly activated compared to *S. cerevisiae*. This is clearly demonstrated by the individual growth curves (Figure 4). Still, one wonders if, when mixed, these two species will show growth dynamics that are consistent with *S. uvarum* winning the competition at high galactose, but may be *S. cerevisiae* winning at low galactose plus glucose (or mannose, fructose)? Or maybe, *S. uvarum's GAL* network is superior across the board (I'd expect tradeoffs, but still)? In general, direct competition may reveal a bit more about the evolutionary contexts under which the *GAL* networks diverged differently in various species, but this just a suggestion, not a request for additional data.

4) In the subsection “*S. uvarum* has two co-repressors with partially overlapping functions”, at the start of the second paragraph: why "although", maybe “because”?

5) Inevitably, jargon and acronyms are used massively, and to some readers (this reviewer, for example) this comprises a hurdle. May be at least add one word to make it easier, e.g. TGA phenotype, or PGM gene.

6) Discussion (subsection “The non-equivalence of sugars in contributing to metabolic overload”, last paragraph): I'm not sure that human metabolic defects are so relevant.

7) Concluding paragraph: I can imagine situations where paralogue loss may also relief clashes (no immediate reference comes to mind, except that in domesticated strains prologue inactivation is quite common, as are WGDs of course).

---

## [Author Response]

The reviewers and the Reviewing Editor agree that no additional experiments are required for the main message of the manuscript, although if data is available that addresses some of the questions raised by the reviewers, the authors should include it. The full reviews are included below as they note a number of points where clarification or modification of claims would help the reader and increase the precision with which the results and conclusions is communicated.

In most cases, we were able to respond to the points raised by modifying the text, reformatting figures, or referencing published studies. In one case, we performed additional experiments so that the strains being compared were grown and analyzed together. Specifically, as suggested by Justin Fay and Reviewer #2, in Figure 2, we included *ScerGAL3* as a control to directly compare it with the replacement of the *ScerGAL1* coding sequence by *SuvaGAL3*.

In particular, please clarify the nature of the strains used (point #2 of reviewer #3).

We have added "when derepressed" to the subheading title of this section to make it clear that the comparisons being made are among co-repressor mutants, "The less active *S. cerevisiae GAL* network is less susceptible to metabolic overload when derepressed".

*And address the relationship between the metabolic changes and the growth phenotypes (end of second paragraph of reviewer #2: what is the evidence that metabolic overload is causal? Does this inference need to be toned down?)*

We agree that we have not shown a strict causal link. We presented several orthogonal forms of evidence that rule out many trivial explanations, and we hope this study will lead to more detailed mechanistic studies in the future. Some claims were overstated in the original manuscript, so we have carefully gone through the entire text to tone down the strength of this inference. Examples are cited in the response to Reviewer #2.

*Reviewer #1:*

*Two sets of experiments make use of precise allele replacements of the GAL1 promoter. These precise replacements are commendable since they avoid certain issues, however they have the disadvantage of GAL10 being driven from the same promoter. Can the GAL1 promoter swaps be attributed to GAL1 or GAL10 activity? While the main message of the work does not hinge on the answer, is there data that can address this or can you point this out to the reader. Replacement of the GAL1 coding region should not be affect by this issue, although there is an interesting lncRNA that covers the GAL cluster and is involved in regulation (PMID: 26833086).*

We completely agree that our *GAL1* promoter swaps also affected the shared *GAL10* divergent promoter, a relevant fact not clearly stated in the original manuscript. We have added a sentence discussing this molecular constraint and the lncRNA, "Since the *GAL1-GAL10* promoter is a divergent promoter, genetic modifications (evolved or engineered) inevitably impact both genes, as well as perhaps a lncRNA previously described in *S. cerevisiae* (Cloutier et al. 2016)." We also modified the Methods, subsection “Strain construction”. Interestingly, evolution has also been constrained by this divergent promoter architecture, and many mutations are expected to have jointly affected *GAL1, GAL10*, and potentially lncRNA expression. It might be an interesting mechanistic or synthetic biology question to ask whether changing *GAL1* expression would be sufficient without affecting *GAL10* or the lncRNA (e.g. by moving it to another locus), but we feel direct manipulation of the locus is the most appropriate test in this case.

Several other types of data specifically report expression from the *GAL1* side of the divergent promoter, showing that *GAL1* expression is affected, even if it is not necessarily the complete story: the previous gene expression results of Caudy et al. 2013 and Roop et al. 2016, our reporter construct data, and our RNA-Seq data.

*The reduced activity of the GAL genes in S. cerevisiae is striking. However, whenever S. cerevisiae is found to have loss of function I always wonder whether this is a defect specific to the lab strain. The lab strain has been found to a have a particularly long lag phase in glucose/galactose mixtures compared to other S. cerevisiae strains which also show less GAL repression as glucose levels fall (PMID: 25626068). While this concern does not require repeating the experiments in another strain background, it should be addressed in the manuscript, preferably through a single experiment: e.g. gal80 deletion.*

The specific experiments suggested (i.e. *gal80* deletions in additional strains of *S. cerevisiae*) could conceivably find that some strains of *S. cerevisiae* had a slightly stronger TGA phenotype than the barely detectable one we report here for the first time. Even so, previously published data and the present data suggest that multiple *S. cerevisiae* strains grow faster in galactose when the co-repressor is removed. Specifically, here we report the *gal80* mutant phenotype of *S. cerevisiae* in the RM11-1a background, a derivative of a vineyard strain (a fact now made explicit in the legend of Figure 4, rather than just being buried in the supplement). Previous observations were made in S288c (Hittinger at al. 2010), W303 (Segre et al. 2006), and 21R (Torchia et al. 1984), a background from Jim Hopper's lab mainly comprised mainly of S288c. Thus, at least four studies and four (partially overlapping) genetic backgrounds have made the observation that *S. cerevisiae gal80* mutants grow faster than wild-type in galactose. *S. kudriavzevii* also shares this *gal80* mutant phenotype (Hittinger et al. 2010), which further suggests that the key difference is not specific to lab strains of *S. cerevisiae*, but is rather a difference between *S. uvarum* and yeasts that diverged later, including at least the *S. kudriavzevii-S. cerevisiae* clade. In the revised manuscript, we explicitly describe the evidence of these additional genetic backgrounds in the first paragraph of the subsection “Strains lacking the co-repressors arrest their growth”.

There is also considerable evidence that the differences in *GAL* network activity and diauxic shift being examined in our study reflect strong species-level differences, rather than the apparently subtler differences between *S. cerevisiae* strains. The Wang et al. 2015 study referenced did indeed observe a range of diauxic shifts among *S. cerevisiae* strains, especially in 0.25% glucose + 0.25% galactose, conditions fairly far from the higher concentration conditions tested in our study. Our conditions were much closer to the conditions of Roop et al. 2016 (1% glucose + 1% galactose) who also observed substantial differences between *S. uvarum* (called *S. bayanus* by these authors) and *S. cerevisiae*. In contrast to Wang et al. 2015, Roop et al. 2016 observed that all *S. cerevisiae* strains examined underwent diauxic shift. Most of the examined strains by these two studies did not overlap, but strain BC187 was examined by both authors and seems highly informative about the differences between their observations. Wang et al. 2015 found BC187 to have a very limited diauxic shift in their conditions, while Roop et al. 2016 found it had a strong diauxic shift in their conditions, conditions where *S. uvarum* still had essentially no diauxic shift. It is not clear whether either condition is ecologically relevant, but each condition is clearly better tuned to revealing quantitative variation at a specific scale.

In summary, we do not doubt that there are experimental conditions where the subtler genetic variation between *S. cerevisiae* strains results in reproducible diauxic shifts, nor that future studies may uncover quantitative variation in their *gal80* mutant phenotypes. Nevertheless, we feel there is sufficient evidence that our key findings (and those of Roop et al. 2016 and Caudy et al. 2013) represent species-level divergence that differs in degree, if not kind, from the variation segregating within *S. cerevisiae*.

*The evidence that SuvaGAL3 encodes a functional galactokinase needs more support for the strength of the statements made: "SuvaGAL3 does indeed encode a functional galactokinase". The Suva gal1 null did not grow on galactose consistent with Suva GAL3 lacking galactokinase activity. However, when ScerGAL1 was replaced with SuvaGAL3 there was growth, indicating sufficient activity for growth because ScerGAL1-promoter is strong. Either biochemical evidence is needed that SuvaGAL3 and ScerGAL3 differ in their activity or the control is needed where ScerGAL3 is also overexpressed by the Scer GAL1 promoter. A last option would be to temper this finding, since it is not essential to the main message.*

We agree that we have not directly tested biochemical activity and have removed the cited sentence making this claim. We have also added "likely" to the Figure 2 legend subtitle.

The suggested control was originally done in Hittinger and Carroll 2007, previous findings not explicitly discussed in the original manuscript. We considered making a table to summarize our prior and current findings, but we felt that the clarity of the manuscript would be substantially strengthened if we redid all the relevant experiments in parallel, as well as ensuring that comparisons were not affected by being performed in different labs a decade apart. This new experiment yielded the expected results and is reported in the revised Figure 2.

In addition to adding this experiment and eliminating the direct claim of biochemical activity, we also reordered this section (subsection “Less partitioned galactokinase and co-induction functions”, first paragraph) to more clearly present the genetic evidence.

*Reviewer #2:*

*This manuscript describes differences between two budding yeast species in the effects of mutations in the galactose metabolic pathway. The experiments are carefully done and together make up a sizeable body of work. The conclusions are for the most part justified, although I note some exceptions below.*

*The bulk of the manuscript deals with the discovery of a transient growth arrest in galactose medium in an S. uvarum strain in which the two paralogs of the GAL80 repressor are deleted. This stands in contrast to the growth advantage in S. cerevisiae upon deletion of its one GAL80 copy (Figure 4). Transcriptomic and metabolomic experiments show that in this S. uvarum double mutant in galactose, during a short burst of rapid growth followed by arrest, glycolysis is upregulated and the TCA cycle is repressed (Figure 5). Deleting GAL1 (Figure 6) or replacing the GAL1 promoter with the S. cerevisiae allele (Figure 7) was sufficient to rescue the transient growth arrest phenotype of the S. uvarum double GAL80 mutant, at least in conditions when galactose was mixed with other sugars. Thus the S. uvarum GAL1 promoter, seemingly by virtue of its high activity (Figure 3), is a driver of the transient growth arrest of the S. uvarum double GAL80 mutant, at least in mixed sugar media. The hyperactivity of the S. uvarum GAL1 promoter is further underlined by the finding that only it, and not the S. cerevisiae allele of the GAL1 promoter, can compensate for the loss of GAL3 in an S. cerevisiae background during growth in galactose (Figure 2). The authors' model is that the transient growth arrest phenotype of the S. uvarum double GAL80 mutant accrues from overzealous galactose metabolism leading to a broader "metabolic overload." Strictly speaking, any mechanistic role for the suite of expression and metabolite changes seen in this double mutant is not shown.*

As noted above in the response to the editor, we agree that we have not made a firm mechanistic link between the TGA phenotype and the suite of expression and metabolic changes. We have carefully gone through the entire manuscript and softened conclusions where needed, including the following:

1) We changed the subheading title "Overactive galactose catabolism causes widespread metabolic and regulatory defects" to "Overactive galactose catabolism precedes widespread metabolic and regulatory defects".

2) In the Abstract, we changed “due to overly rapid galactose catabolism” to “likely due to overly rapid galactose catabolism”

3) In the third paragraph of the subsection “Overactive galactose catabolism precedes widespread metabolic and regulatory defects”, we changed "was insufficient to explain TGA" to "seemed unlikely to be sufficient to explain the TGA phenotype."

4) At the end of the subsection “Specific sugars can exacerbate metabolic overload”, we changed "show" to "suggest".

5) In the subsection “Biodiversity offers a panoramic window to molecular biology”, we removed “Metabolic overload led to the accumulation of […] decoupled from gene regulation”, rephrasing this to “[…] likely due to metabolic overload […]”.

6) In the last paragraph of the subsection “Strains lacking the co-repressors arrest their growth”, we changed "caused by" to "associated with"

7) In the third paragraph of the subsection “Overactive galactose catabolism precedes widespread metabolic and regulatory defects”, we changed "To further test the hypothesis that overly rapid galactose catabolism caused TGA” to " To further characterize how overly rapid galactose catabolism might lead to the TGA phenotype, […]".

8) At the end of the subsection “Overactive galactose catabolism precedes widespread metabolic and regulatory defects”, we added "likely".

*The authors make the interesting argument that the second paralog of GAL80, GAL80b, may have evolved in S. uvarum as a necessary brake on its highly active galactose metabolic program. Indeed, they show that GAL80b in S. uvarum is a stronger repressor of GAL1 than is GAL80 (Figure 3). More support for fitness-relevant functions of GAL80b could strengthen the manuscript. Since GAL80 alone is sufficient to prevent the transient growth arrest that otherwise manifests in the S. uvarum double GAL80 mutant in pure galactose (Figure 4—figure supplement 1), any benefit of GAL80b wouldn't be in this condition. Rather, the GAL80b mutant in S. uvarum has a slight growth defect in a mixed sugar environment*

First, we note that our evidence shows *GAL80B* is playing a more important role in repression in specific mixed sugar conditions. Arriving at conditions sensitive enough to detect a defect was challenging, and previous researchers were not successful (Caudy et al. 2013). The fact that we can only detect a growth defect in some conditions does not necessarily mean that it is not playing an evolutionarily important role in other conditions. Laboratory experiments lack the precision and sensitivity of natural selection, which can select against mutations whose growth defects are approximately the inverse of the effective population size.

*(Figure 3; it would be good to show the GAL80 mutant also in this figure).*

The key finding is that *GAL80B* is required for a wild-type growth rate in at least one condition, a condition where a second braking mechanism is beneficial. If *gal80* mutants also had a quantitative effect in this condition, it is not clear that this result would change this conclusion (unfortunately, for this reason, that potentially interesting experiment has not been done). We note that Figure 3 shows that *gal80b* mutants have more of an effect on *GAL1* reporter expression than *gal80* mutants in mixed sugar conditions.

*The mechanism of the latter phenotype would be of interest here, particularly in relation to S. uvarum's hyperactive galactose metabolic program which is the theme of the work.*

Our working hypothesis is that the prominence of *GAL80B* in these conditions is largely due to its higher expression in the presence of galactose, rather than stronger protein activity per se. We have rephrased the last paragraph of the subsection “*S. uvarum* has two co-repressors with partially overlapping functions” to help highlight this view, "Perhaps because of its dynamic expression, the deletion mutant phenotype of *S. uvarum gal80b∆* proved condition dependent." We agree that it is too preliminary to firmly conclude anything about this mechanism, but it would make for an interesting future study.

*A first part of the manuscript, of relatively modest scope, focuses on GAL3, which is a regulator in S. cerevisiae. Figure 2 shows that S. cerevisiae can't grow in galactose when GAL1 is deleted (an S. uvarum GAL1 mutant is not reported).*

As in *S. cerevisiae*, the *S. uvarum gal1* mutant does not grow on galactose. This data is reported in Figure 2—figure supplement 1 and discussed in the first paragraph of the subsection “Less partitioned galactokinase and co-induction functions”, which has been rewritten to improve clarity.

*This defect can be rescued by the S. uvarum allele of GAL3, indicating a gain of metabolic function by the latter which the authors infer to be galactokinase activity.*

As discussed above in the response to Justin Fay, we have added additional experiments to Figure 2 and removed the conclusion that galactokinase activity was demonstrated directly. Based on prior work in *K. lactis*, we also note that *S. cerevisiae GAL3* probably lost metabolic function, rather than it being gained by *S. uvarum GAL3*.

*Reviewer #3:*

*The paper is clearly written, and despite technical complexity and detail-richness, makes an interesting read. The conclusions are in my view well supported, and with adequate controls that use dot be a standard but are now missing in many studies (e.g. reintroduction of the knocked out gene in a different locus to ensure the phenotype is related to this gene missing and not some other genomic perturbation). The study addressed multiple aspects including changes in transcriptome and metabolomics (including the identification of galactose-1-phosphate as the origins of toxicity). While I could imagine (and could not resist suggesting) interesting experiments that may provide further insights, I am impressed by the breadth of the data and there is no need in my view to demand additional experiments.*

We thank the reviewer for their supportive comments and appreciation of this work. We note that, although galactose-1-phosphate toxicity could be playing a minor role, metabolic defects downstream of this step are much more substantial.

*Comments:*

*1) GAL gene content and sequence differences:*

*Reading the next section one wonders if the S. uvarum Gal1/3 the products of the WGD? But then, the authors state: that divergence of "ScerGAL1 and ScerGAL3 happened after the divergence of S. uvarum and S. cerevisiae. I suggest to explicitly discuss Gal1/3 in this section. Further, although these are not the results of this study, still, a schematic phylogenetic tree(s) for the species and genes addressed here would really help the reader who is not familiar with yeast phylogeny and/or the Gal system.*

In the subsection “GAL gene content and sequence differences” it now clearly states that *GAL1* and *GAL3* are paralogs generated by the WGD. We have also included the explanation for the inference at end of the next section, “Thus, it is likely that the homologs in the common ancestor of *S. uvarum* and *S. cerevisiae* were more functionally redundant than in modern *S. cerevisiae* […]” Since these genes started to functionally diverge well before the origin of the genus *Saccharomyces*, we have softened the conclusion about *GAL1* and *GAL3* divergence, so that it reads, "considerable subfunctionalization", instead of "much of the subfunctionalization" We considered moving Figure 8, which contains the schematic phylogenetic trees, to earlier in the study; unfortunately, the main purpose of this figure is to summarize and discuss the results of the study. We hope that the added text clarifying the relationships of *GAL1* and *GAL3* and the final layout of the manuscript will make the information sufficiently accessible to readers.

*2) "The less active S. cerevisiae GAL network is less susceptible to metabolic overload" The data in this section relates to the GAL80 knockouts of both species. Wouldn't the right comparison be the wild-types?*

As discussed in the response to the editor, we have changed the title of this section by adding "when derepressed" to better reflect the results. We have only observed the TGA phenotype and metabolic overload in mutants; it is likely that the wild-type gene networks of both species do a decent job of preventing metabolic overload under normal conditions. The means by which the species achieve this regulation is what differs: *S. cerevisiae* has a low-activity network and less robust repression, while *S. uvarum* has a high-activity network and more robust repression.

*3) Related to the above: Galactose catabolism in S. uvarum can be highly activated compared to S. cerevisiae. This is clearly demonstrated by the individual growth curves (Figure 4). Still, one wonders if, when mixed, these two species will show growth dynamics that are consistent with S. uvarum winning the competition at high galactose, but may be S. cerevisiae winning at low galactose plus glucose (or mannose, fructose)? Or maybe, S. uvarum's GAL network is superior across the board (I'd expect tradeoffs, but still)? In general, direct competition may reveal a bit more about the evolutionary contexts under which the GAL networks diverged differently in various species, but this just a suggestion, not a request for additional data.*

Direct competition experiments could prove much more sensitive for detecting some of the tradeoffs that the reviewer hypothesizes, and we agree it may be worth exploring this system further in that context. Growth curve analyses are a much less sensitive technique, so the differences that we do report are quite large indeed. Thus, the conclusions we have reached are well supported, even if there might be additional differences that we lacked sensitivity to detect. Even with this limitation, growth curves have the added benefit of being able to study the culture dynamics of an individual strain, which was key to characterizing the previously overlooked TGA phenotype.

*4) In the subsection “S. uvarum has two co-repressors with partially overlapping functions”, at the start of the second paragraph: why "although", maybe “because”?*

We agree that "Although" did not quite convey our meaning, and we have rephrased this to explicitly focus on the gene expression differences between the paralogs: " Perhaps because of its dynamic expression, […]".

*5) Inevitably, jargon and acronyms are used massively, and to some readers (this reviewer, for example) this comprises a hurdle. May be at least add one word to make it easier, e.g. TGA phenotype, or PGM gene.*

This is an excellent suggestion, and we have added "phenotype" or "phase" after each reference to TGA. We have also added "gene" to uses of *PGM1* where it might have been unclear from context. Most of the genes (e.g. *GAL1, GAL3, GAL80, GAL80B*) or the proteins they encode were referred to often enough in the study that we trust the readers will be able to follow it, but we welcome any other specific suggestions.

6) Discussion (subsection “The non-equivalence of sugars in contributing to metabolic overload”, last paragraph): I'm not sure that human metabolic defects are so relevant.

We have deleted most of this paragraph. We also shortened the previous paragraph to make room for a very brief mention of the reported link between fructose overconsumption and cancer and diabetes (subsection “The non-equivalence of sugars in contributing to metabolic overload”).

*7) Concluding paragraph: I can imagine situations where paralogue loss may also relief clashes (no immediate reference comes to mind, except that in domesticated strains prologue inactivation is quite common, as are WGDs of course).*

We agree that this is a very interesting idea worthy of further investigation. We could not think of a precise example either, and we do not think our system meets the criteria, at least with current data. To at least advertise this hypothesis, we now refer to the recently proposed idea of "paralog interference" in the last paragraph of the subsection “Ongoing functional diversification of paralogs and their gene networks”. We reference a study from Sandy Johnson's lab that proposed specific amino acid substitutions as a way of relieving paralog interference, but one could imagine paralog loss as being another way of accomplishing this relief in certain circumstances.